# Segmenting Watermarked Texts From Language Models

**Xingchi Li**[*]
Department of Statistics
Texas A&M University
College Station, TX 77843
anthony.li@stat.tamu.edu

**Guanxun Li**[*]
Department of Statistics
Beijing Normal University at Zhuhai
Zhuhai, Guangdong 519087
guanxun@bnu.edu.cn

**Xianyang Zhang**[†]
Department of Statistics
Texas A&M University
College Station, TX 77843
zhangxiany@stat.tamu.edu

## Abstract

Watermarking is a technique that involves embedding nearly unnoticeable statistical signals within generated content to help trace its source. This work focuses on a scenario where an untrusted third-party user sends prompts to a trusted language model (LLM) provider, who then generates a text from their LLM with a watermark. This setup makes it possible for a detector to later identify the source of the text if the user publishes it. The user can modify the generated text by substitutions, insertions, or deletions. Our objective is to develop a statistical method to detect if a published text is LLM-generated from the perspective of a detector. We further propose a methodology to segment the published text into watermarked and non-watermarked sub-strings. The proposed approach is built upon randomization tests and change point detection techniques. We demonstrate that our method ensures Type I and Type II error control and can accurately identify watermarked sub-strings by finding the corresponding change point locations. To validate our technique, we apply it to texts generated by several language models with prompts extracted from Google's C4 dataset and obtain encouraging numerical results.[1]

## 1 Introduction

With the increasing use of large language models in recent years, it has become essential to differentiate between text generated by these models and text written by humans. Some of the most advanced LLMs, such as GPT-4, Llama 3, and Gemini, are very good at producing human-like texts, which could be challenging to distinguish from human-generated texts, even for humans. However, it is crucial to distinguish between human-produced texts and machine-produced texts to prevent the spread of misleading information, improper use of LLM-based tools in education, model extraction attacks through distillation, and the contamination of training datasets for future language models.

Watermarking is a principled method for embedding nearly unnoticeable statistical signals into text generated by LLMs, enabling provable detection of LLM-generated content from its human-written

---

[*]Equal contribution

[†]Corresponding author

[1]We release all code publicly at `https://github.com/doccstat/llm-watermark-cpd`.

38th Conference on Neural Information Processing Systems (NeurIPS 2024).

counterpart. This work focuses on a scenario where an untrusted third-party user sends prompts to a trusted large language model (LLM) provider, who then generates a text from their LLM with a watermark. This makes it possible for a detector to later identify the source of the text if the user publishes it. The user is allowed to modify the generated text by making substitutions, insertions, or deletions before publishing it. We aim to develop a statistical method to detect if a published text is LLM-generated from the perspective of a detector. When the text is declared to be watermarked, we study a problem that has been relatively less investigated in the literature: to separate the published text into watermarked and non-watermarked sub-strings. In particular, we divide the texts into a sequence of moving sub-strings with a pre-determined length and sequentially test if each sub-string is watermarked based on the $p$-values obtained from a randomization test. Given the $p$-value sequence, we examine if there are structural breaks in the underlying distributions. For non-watermarked sub-strings, the $p$-values are expected to be uniformly distributed over $[0, 1]$, while for watermarked sub-strings, the corresponding $p$-values are more likely to concentrate around zero. By identifying the change points in the distributions of the $p$-values, we can segment the texts into watermarked and non-watermarked sub-strings.

In theory, we demonstrate that our method ensures Type I and Type II error control and can accurately identify watermarked sub-strings by finding the corresponding change point locations. Specifically, we obtain the convergence rate for the estimated change point locations. To validate our technique, we apply it to texts generated by different language models with prompts extracted from the real news-like data set from Google's C4 dataset, followed by different modifications/attacks that can introduce change points in the text.

## 1.1 Related works and contributions

Digital watermarking is a field that focuses on embedding a signal in a medium, such as an image or text, and detecting it using an algorithm. Early watermarking schemes for natural language processing-generated text were presented by Atallah et al. [2001]. Hopper et al. [2007] formally defined watermarking and its desired properties, but their definitions were not specifically tailored to LLMs. More recently, Abdelnabi and Fritz [2021] and Munyer and Zhong [2023] proposed schemes that use machine learning models in the watermarking algorithm. These schemes take a passage of text and use a model to produce a semantically similar altered passage. However, there is no formal guarantee for generating watermarked text with desirable properties when using machine learning.

The current work is more related to Aaronson [2023], Kirchenbauer et al. [2023a], Kuditipudi et al. [2023]. In particular, Kirchenbauer et al. [2023a] introduced a "red-green list" watermarking technique that splits the vocabulary into two lists based on hash values of previous tokens. This technique slightly increases the probability of embedding the watermark into "green" tokens. Other papers that discuss this type of watermarking include Kirchenbauer et al. [2023b], Cai et al. [2024], Liu and Bu [2024], and Zhao et al. [2023]. However, this watermarking technique is biased, as the next token prediction (NTP) distributions have been modified, leading to a performance degradation of the LLM. Aaronson [2023] describes a technique for watermarking LLMs using exponential minimum sampling to sample tokens from an LLM, where the inputs to the sampling mechanism are also a hash of the previous tokens. This approach is closely related to the so-called Gumbel trick in machine learning. Kuditipudi et al. [2023] proposed an inverse transform watermarking method that can be made robust against potential random edits. Other unbiased watermarks in this fast-growing line of research include Zhao et al. [2024], Fernandez et al. [2023], Hu et al. [2023], Wu et al. [2023].

However, less attention has been paid to understanding the statistical properties of watermark generation and detection schemes. The paper by Huang et al. [2023] considers watermark detection as a problem of composite dependence testing. The authors aim to understand the minimax Type II error and the most powerful test that achieves it. However, Huang et al. [2023] assumes that the NTP distributions remain unchanged from predicting the first token to the last token, which can be unrealistic. On the other hand, Li et al. [2024] introduced a flexible framework for determining the statistical efficiency of watermarks and designing powerful detection rules. This framework reduces the problem of determining the optimal detection rule to solving a minimax optimization program.

Compared to the existing literature, we make two contributions:
(a) We rigorously study the Type I and Type II errors of the randomization test to test the existence of a watermark; see Theorems 1-2. We apply these results to the recently proposed inverse transform watermark and Gumbel watermark schemes; see Corollary 1.

(b) We develop a systematic statistical method to segment texts into watermarked and non-watermarked sub-strings. We have also investigated the theoretical and finite sample performance of this methodology. As far as we know, this problem has not been studied in recent literature on generating and detecting watermarked texts from LLMs.

## 2 Problem setup

### 2.1 Watermarked text generation

Denote by $\mathcal{V}$ the vocabulary (a discrete set), and let $P$ be an autoregressive LLM which maps a string $y_{-n_0:t-1} = y_{-n_0} y_{-n_0+1} \cdots y_{t-1} \in \mathcal{V}^{t+n_0}$ to a distribution over the vocabulary, with $p(\cdot|y_{-n_0:t-1})$ being the distribution of the next token $y_t$. Here $y_{-n_0:0}$ denotes the prompt provided by the user. Set $V = |\mathcal{V}|$ be the vocabulary size, and let $\xi_{1:t} = \xi_1 \xi_2 \cdots \xi_t$ be a watermark key sequence with $\xi_i \in \Xi$ for each $i$, where $\Xi$ is a general space. Given a prompt sent from a third-party user, the LLM provider calls a generator to autoregressitvely generate text from an LLM using a decoder function $\Gamma$, which maps $\xi_t$ and a distribution $p_t$ over the next token to a value in $\mathcal{V}$. The watermarking scheme should preserve the original text distribution, i.e., $P(\Gamma(\xi_t, p_t) = y) = p_t(y)$. A watermark text generation algorithm recursively generates a string $y_{1:n}$ by

$$y_i = \Gamma(\xi_i, p(\cdot|y_{-n_0:i-1})), \quad 1 \le i \le n,$$

where $n$ is the number of tokens in the text $y_{1:n}$ generated by the LLM, and $\xi_i$'s are assumed to be independently generated from some distribution $\nu$ over $\Xi$. In other words, given $p(\cdot|y_{-n_0:i-1})$, $y_i$ is completely determined by $\xi_i$ and $y_{-n_0:i-1}$.

For ease of presentation, we associate each token in the vocabulary with a unique index from $[V] := \{1, 2, \ldots, V\}$, and we remark that the generator should be invariant to this assignment.

**Example 1** (Inverse transform sampling). In this example, we describe a watermarked text generation method developed in Kuditipudi et al. [2023] and discuss an alternative formulation of their approach. Write $\mu_i(k) = p(k|y_{-n_0:i-1})$ for $1 \le k \le V$ and $1 \le i \le n$. To generate the $i$th token, we consider a permutation $\pi_i$ on $[V]$ and $u_i \sim \text{Unif}[0,1]$ (which denotes the uniform distribution over $[0,1]$), which jointly act as the key $\xi_i$. Let

$$\Gamma((\pi_i, u_i), \mu_i) = \pi_i^{-1}(\min\{\pi_i(l) : \mu_i(j : \pi_i(j) \le \pi_i(l)) \ge u_i\}).$$

We note that $\Gamma((\pi_i, u_i), \mu_i) = k$ if $\min\{\pi_i(l) : \mu_i(j : \pi_i(j) \le \pi_i(l)) \ge u_i\} = \pi_i(k)$, which implies that

$$\mu_i(j : \pi_i(j) \le \pi_i(k)) \ge u_i > \mu_i(j : \pi_i(j) < \pi_i(k)).$$

As the length of this interval is $\mu_i(k)$, $P(\Gamma((\pi_i, u_i), \mu_i) = k) = \mu_i(k)$. An alternate way to describe the same generator is as follows: Given a partition of the interval $[0,1]$ into $V$ sub-intervals denoted by $\mathcal{I}_1, \ldots, \mathcal{I}_V$, we can order them in such a way that each interval $\mathcal{I}_i$ is adjacent to its immediate right neighbor $\mathcal{I}_{i+1}$. Now let $\mathcal{I}(k; i)$'s be $V$ intervals with the length $|\mathcal{I}(k; i)| = p(k|y_{-n_0:i-1})$ for $1 \le k \le V$. Given the permutation $\pi_i$, we can order the $V$ intervals through $\pi_i$, i.e., $\{\mathcal{I}_{\pi_i(k)}(k; i) : k = 1, 2, \ldots, V\}$. Define $\xi_{ik} = \mathbf{I}\{u_i \in \mathcal{I}_{\pi_i(k)}(k; i)\}$ and set $y_i = k$ if $\xi_{ik} = 1$. Clearly $P(y_i = k) = P(\xi_{ik} = 1) = P(u_i \in \mathcal{I}_{\pi_i(k)}(k; i)) = p(k|y_{-n_0:i-1})$.

Now, for a string $\widetilde{y}_{1:n}$ (with the same length as the key) that is possibly watermarked, Kuditipudi et al. [2023] suggest the following metric to quantify the dependence between the watermark key and the string:

$$\mathcal{M}(\xi_{1:n}, \widetilde{y}_{1:n}) = \frac{1}{n} \sum_{i=1}^{n} (u_i - 1/2) \left( \frac{\pi_i(\widetilde{y}_i) - 1}{V - 1} - \frac{1}{2} \right). \tag{1}$$

Observe that if $\widetilde{y}_i$ is generated using the above scheme with the key $\xi_i = (\pi_i, u_i)$, then $u_i$ and $\pi_i(\widetilde{y}_i)$ are positively correlated. Thus, a large value of $\mathcal{M}$ indicates that $\widetilde{y}_{1:n}$ is potentially watermarked.

**Example 2** (Exponential minimum sampling). We describe another watermarking technique proposed in Aaronson [2023]. To generate each token of a text, we first sample $\xi_{ik} \sim \text{Unif}[0,1]$ independently for $1 \le k \le V$. Let

$$y_i = \underset{1 \le k \le V}{\arg\max} \frac{\log(\xi_{ik})}{p(k|y_{-n_0:i-1})} = \underset{1 \le k \le V}{\arg\min} \frac{-\log(\xi_{ik})}{p(k|y_{-n_0:i-1})} = \underset{1 \le k \le V}{\arg\min} E_{ik},$$

where $E_{ik} := -\log(\xi_{ik})/p(k|y_{-n_0:i-1}) \sim \text{Exp}(p(k|y_{-n_0:i-1}))$ with $\text{Exp}(a)$ denoting an exponential random variable with the rate $a$. For two exponential random variables $X \sim \text{Exp}(a)$ and $Y \sim \text{Exp}(b)$, we have two basic properties: (i) $\min(X, Y) \sim \text{Exp}(a + b)$; (ii) $P(X < Y) = \mathbb{E}[1 - \exp(-aY)] = a/(a + b)$. Using (i) and (ii), it is straightforward to verify that

$$P(y_i = k) = P\left(E_{ik} < \min_{j \neq k} E_{ij}\right) = p(k|y_{-n_0:i-1}).$$

Hence, this generation scheme preserves the original text distribution.

Aaronson [2023] proposed to measure the dependence between a string $\widetilde{y}_{1:n}$ and the key sequence $\xi_{1:n}$ using the metric

$$\mathcal{M}(\xi_{1:n}, \widetilde{y}_{1:n}) = \frac{1}{n} \sum_{i=1}^{n} \{\log(\xi_{i,\widetilde{y}_i}) + 1\}. \tag{2}$$

The idea behind the definition of this function is that if $\widetilde{y}_i$ was generated using the key $\xi_i$, then $\xi_{i,\widetilde{y}_i}$ tends to have a higher value than the other components of $\xi_i$. Therefore, a larger value of the metric indicates that the string $\widetilde{y}_{1:n}$ is more likely to be watermarked.

## 2.2 Watermarked text detection

We now consider the detection problem, which involves determining whether a given text is watermarked or not. Consider the case where a string $\widetilde{y}_{1:m}$ is published by the third-party user and a key sequence $\xi_{1:n}$ is provided to a detector. The detector calls a detection method to test

$$H_0 : \widetilde{y}_{1:m} \text{ is not watermarked} \quad \text{versus} \quad H_a : \widetilde{y}_{1:m} \text{ is watermarked},$$

by computing a $p$-value with respect to a test statistic $\phi(\xi_{1:n}, \widetilde{y}_{1:m})$. It is important to note that the text published by the user can be quite different from the text initially generated by the LLM using the key $\xi_{1:n}$, which we refer to as $y_{1:n}$. To account for this difference, we can use a transformation function $\mathcal{E}$ that takes $y_{1:n}$ as the input and produces the published text $\widetilde{y}_{1:m}$ as the output, i.e., $\widetilde{y}_{1:m} = \mathcal{E}(y_{1:n})$. This transformation can involve substitutions, insertions, deletions, paraphrases, or other edits to the input text.

The test statistic $\phi$ measures the dependence between the text $\widetilde{y}_{1:m}$ and the key sequence $\xi_{1:n}$. Throughout our discussions, we will assume that a large value of $\phi$ provides evidence against the null hypothesis (e.g., stronger dependence between $\widetilde{y}_{1:m}$ and $\xi_{1:n}$). To obtain the $p$-value, we consider a randomization test. In particular, we generate $\xi_i^{(t)} \sim \nu$ independently over $1 \leq i \leq n$ and $1 \leq t \leq T$, and $\xi_i^{(t)}$s are independent with $\widetilde{y}_{1:m}$. Then the randomization-based $p$-value is given by

$$p_T = \frac{1}{T+1}\left(1 + \sum_{t=1}^{T} \mathbf{1}\{\phi(\xi_{1:n}, \widetilde{y}_{1:m}) \leq \phi(\xi_{1:n}^{(t)}, \widetilde{y}_{1:m})\}\right).$$

**Theorem 1.** For the randomization test, we have the following results.

(i) Under the null, $P(p_T \leq \alpha) = \lfloor(T+1)\alpha\rfloor/(T+1) \leq \alpha$, where $\lfloor a \rfloor$ denotes the greatest integer that is less than or equal to $a$;

(ii) Suppose the following three conditions hold:

(a) $\max\{\text{Var}(\phi(\xi_{1:n}, \widetilde{y}_{1:m})|\mathcal{F}_m), \text{Var}(\phi(\xi'_{1:n}, \widetilde{y}_{1:m})|\mathcal{F}_m)\} \leq C_v/n$ with $C_v > 0$;

(b) $\mathbb{E}[\phi(\xi'_{1:n}, \widetilde{y}_{1:m})|\mathcal{F}_m] = O(n^{-1/2})$;

(c) $\lim_{n \to \infty} \sqrt{n}\mathbb{E}[\phi(\xi_{1:n}, \widetilde{y}_{1:m})|\mathcal{F}_m] = \infty$.

Here $\mathcal{F}_m = [y_{-n_0:0}, \widetilde{y}_{1:m}]$ and $\xi'_{1:n}$ is a key sequence generated in the same way as $\xi_{1:n}$ but is independent of $\widetilde{y}_{1:m}$. Given any $\epsilon > 0$, when $T > 2/\epsilon - 1$,

$$P(p_T \leq \alpha|\mathcal{F}_m) \geq 1 - C_1 \exp(-2T\epsilon^2) + o(1), \tag{3}$$

as $n \to +\infty$, where $C_1 > 0$.

We now apply the results in Theorem 1 to Examples 1-2 with $m = n$ and $\phi(\xi_{1:n}, \widetilde{y}_{1:n}) = \mathcal{M}(\xi_{1:n}, \widetilde{y}_{1:n})$ for $\mathcal{M}$ defined in (1) and (2).

**Corollary 1.** *If $\widetilde{y}_{1:n} = y_{1:n}$ and*

$$\frac{1}{\sqrt{n}} \sum_{i=1}^{n} \big(1 - p(y_i | y_{-n_0:i-1})\big) \to \infty, \tag{4}$$

*then Conditions (a)-(c) in Theorem 1 are satisfied for Examples 1-2. Consequently, the power of the randomization test converges to 1 in these examples as $T \to +\infty$.*

However, as the published text can be modified, it is not expected that every token in $\widetilde{y}_{1:m}$ will be related to the key sequence. Instead, we expect certain sub-strings of $\widetilde{y}_{1:m}$ to be correlated with the key sequence under the alternative hypothesis $H_a$. To measure the dependence, we use a scanning method that looks at every segment/sub-string of $\widetilde{y}_{1:m}$ and a segment of $\xi_{1:n}$ with the same length $B$. We use a measure $\mathcal{M}(\xi_{a:a+B-1}, \widetilde{y}_{b:b+B-1})$ to quantify the dependence between $\xi_{a:a+B-1}$ and $\widetilde{y}_{b:b+B-1}$, chosen based on the watermarked text generation method described above. Given $\mathcal{M}$ and the block size $B$, we can define the maximum test statistic as

$$\phi(\xi_{1:n}, \widetilde{y}_{1:m}) = \max_{1 \le a \le n-B+1} \max_{1 \le b \le m-B+1} \mathcal{M}(\xi_{a:a+B-1}, \widetilde{y}_{b:b+B-1}). \tag{5}$$

**Theorem 2.** Consider the maximum statistic defined in (5), where the dependence measure takes the form of $\mathcal{M}(\xi_{a:a+B-1}, \widetilde{y}_{b:b+B-1}) = B^{-1} \sum_{i=0}^{B-1} h_i(\xi_{a+i}, \widetilde{y}_{b+i})$ and $h_i$s are independent conditional on $[\widetilde{y}_{1:m}, y_{-n_0:n}]$. Under the setting of Example 1, $\max_i |h_i| \le 1/4$. In this case, suppose

$$C_{N,B}^{-1} \max_{a,b} \mathbb{E}[\mathcal{M}(\xi_{a:a+B-1}, \widetilde{y}_{b:b+B-1}) | \widetilde{y}_{1:m}, y_{-n_0:n}] \to +\infty, \tag{6}$$

where $N = \max\{n, m\}$ and $C_{N,B} = \sqrt{\log(N)/B}$. Then, (3) holds true. Under the setting of Example 2, $h_i$s are exponentially distributed conditional on $[\widetilde{y}_{1:m}, y_{-n_0:n}]$. In this case, (3) is still true under (6) with $C_{N,B} = \log(N)/B$.

## 3 Sub-string identification

In this section, we aim to address the following question, which seems less explored in the existing literature: given that the global null hypothesis $H_0$ is rejected, how can we identify the sub-strings from the modified text $\widetilde{y}_{1:m}$ that are machine-generated?

To describe the setup, we suppose the text published by the third-party user has the following structure:

$$\underbrace{\widetilde{y}_1 \widetilde{y}_2 \cdots \widetilde{y}_{\tau_1}}_{\text{non-watermarked}} \underbrace{\widetilde{y}_{\tau_1+1} \cdots \widetilde{y}_{\tau_2}}_{\text{watermarked}} \underbrace{\widetilde{y}_{\tau_2+1} \cdots \widetilde{y}_{\tau_3}}_{\text{non-watermarked}} \underbrace{\widetilde{y}_{\tau_3+1} \cdots \widetilde{y}_{\tau_4}}_{\text{watermarked}} \cdots, \tag{7}$$

in which case the sub-strings $\widetilde{y}_{\tau_1+1} \cdots \widetilde{y}_{\tau_2}$ and $\widetilde{y}_{\tau_3+1} \cdots \widetilde{y}_{\tau_4}$ are watermarked. We emphasize that the orders of the watermarked and non-watermarked sub-strings can be arbitrary and do not affect our method described below. The goal here is to separate the text into watermarked and non-watermarked sub-strings accurately. Our key insight to tackling this problem is translating it into a change point detection problem. To describe our method, we define a sequence of moving windows $\mathcal{I}_i = [(i - B/2) \vee 1, (i + B/2) \wedge m]$ with $B$ being the window size which is assumed to be an even number (for the ease of presentation) and $1 \le i \le m$. For each sub-string, we define a randomization-based $p$-value given by

$$p_i = \frac{1}{T+1} \left( 1 + \sum_{t=1}^{T} \mathbf{1}\{\phi(\xi_{1:n}, \widetilde{y}_{\mathcal{I}_i}) \le \phi(\xi_{1:n}^{(t)}, \widetilde{y}_{\mathcal{I}_i})\} \right), \quad 1 \le i \le m, \tag{8}$$

where we let $\phi(\xi_{1:n}, \widetilde{y}_{\mathcal{I}_i}) = \max_{1 \le k \le n} \mathcal{M}(\xi_{\mathcal{J}_k}, \widetilde{y}_{\mathcal{I}_i})$ with $\mathcal{J}_k = [(k - B/2) \vee 1, (k + B/2) \wedge n]$.

We have now transformed the text into a sequence of $p$-values: $p_1, \ldots, p_m$. Under the null, the $p$-values are roughly uniformly distributed, while under the alternatives, the $p$-values will concentrate around zero. Consider a simple setup where the published text can be divided into two halves, with the first half watermarked and the second half non-watermarked (or vice versa). Then, we can identify the change point location through

$$\hat{\tau} = \arg\max_{1 \le \tau < m} S_{1:m}(\tau), \quad S_{1:m}(\tau) := \sup_{t \in [0,1]} \frac{\tau(m - \tau)}{m^{3/2}} |F_{1:\tau}(t) - F_{\tau+1:m}(t)|, \tag{9}$$

where $F_{a:b}(t)$ denotes the empirical cumulative distribution function (cdf) of $\{p_i\}_{i=a}^b$.

We shall use the block bootstrap-based approach to determine if $\hat{\tau}$ is statistically significant. Note that conditional on $\mathcal{F}_m$, the $p$-value sequence is $B$-dependent in the sense that $p_i$ and $p_j$ are independent only if $|i - j| > B$. Hence, the usual bootstrap or permutation methods for independent data may not work as they fail to capture the dependence from neighboring $p$-values. Instead, we can employ the so-called moving block bootstrap for time series data [Kunsch, 1989, Liu et al., 1992]. Given a block size, say $B'$, we create $m - B' + 1$ blocks given by $\{p_i, \ldots, p_{i+B'-1}\}$ for $1 \leq i \leq m - B' + 1$. We randomly sample $m/B'$ (assuming that $m/B'$ is an integer for simplicity) blocks with replacement and paste them together. Denote the resulting resampled $p$-values by $p_1^*, \ldots, p_m^*$. We then compute $F_{a,b}^*(t)$ based on the bootstrapped $p$-values and define

$$S_{1:m}^*(\tau) = \sup_{t \in [0,1]} \frac{\tau(m - \tau)}{m^{3/2}} |F_{1:\tau}^*(t) - F_{\tau+1:m}^*(t)|.$$

Repeat this procedure $T'$ times and denote the statistics by $S_{1:m}^{*,(t)}(\tau)$ for $t = 1, 2, \ldots, T'$. Define the corresponding bootstrap-based $p$-value as

$$\tilde{p}_{T'} = \frac{1}{T'+1} \left( 1 + \sum_{t=1}^{T'} \mathbf{1} \left\{ \max_{1 \leq \tau < m} S_{1:m}(\tau) \leq \max_{1 \leq \tau < m} S_{1:m}^{*,(t)}(\tau) \right\} \right). \tag{10}$$

We claim that there is a statistically significant change point if $\tilde{p}_{T'} \leq \alpha$.

**Remark 1.** Alternatively, one can consider the Cramér-von Mises type statistic $\max_\tau C_{1:m}(\tau)$ with

$$C_{1:m}(\tau) = \int_0^1 \frac{\tau^2(m - \tau)^2}{m^3} |F_{1:\tau}(t) - F_{\tau+1:m}(t)|^2 w(t) dt,$$

to examine the existence of a change point, where $w(\cdot)$ is a non-negative weight function defined over $[0, 1]$. If a change point exists, we can estimate its location through $\tilde{\tau} = \arg\max_{1 \leq \tau < m} C_{1:m}(\tau)$. Other methods to quantify the distributional shift for change point detection include the distance and kernel-based two-sample metrics [Matteson and James, 2014, Chakraborty and Zhang, 2021] and graph-based test statistics [Chen and Zhang, 2015].

Consider the case where there is a single change point located at $\tau^*$. Without loss of generality, let us assume that before the change, the $p$-values are uniformly distributed, i.e., $p_1, \ldots, p_{\tau^*} \sim F_0$ with $F_0(t) = t$ (the cdf of Unif$[0, 1]$), while after the change, the $p$-values follow different distributions that concentrate around zero. We assume that $\liminf_{m \to +\infty} D(F_0, \mathbb{E}[F_{\tau^*+1:m}(t)]) > 0$, where $D(F_0, \mathbb{E}[F_{\tau^*+1:m}(t)]) := \sup_{t \in [0,1]} |F_0(t) - \mathbb{E}[F_{\tau^*+1:m}(t)]|$ is the Kolmogorov-Smirnov distance. In other words, the empirical cdf of the $p$-values from the watermarked sub-strings converges to a distribution that is different from the uniform distribution. This mild assumption allows for the detection of the structure break. As discussed before, we consider the scan statistic $\max_{1 \leq \tau < m} S_{1:m}(\tau)$ with $S_{1:m}(\tau)$ defined in (9) for examining the existence of a change point.

**Proposition 1.** *Assuming that $B \to +\infty$ and $B/m \to 0$, we have*

$$\max_{1 \leq \tau < m} S_{1:m}(\tau) \geq \sqrt{m} \gamma^* (1 - \gamma^*) D(F_0, \mathbb{E}[F_{\tau^*+1:m}(t)]) + o_p(1) \to +\infty,$$

*where $\gamma^* = \lim_{m \to +\infty} \tau^*/m \in (0, 1)$.*

Next, we shall establish the consistency of the change point estimator $\hat{\tau} = \arg\max_{1 \leq \tau < m} S_{1:m}(\tau)$. In particular, we obtain the following result, which establishes the convergence rate of the change point estimate.

**Theorem 3.** *Under the Assumptions in Proposition 1, we have*

$$|\hat{\tau} - \tau^*| = O_p \left( \frac{\sqrt{mB \log(m/B)}}{D(F_0, \mathbb{E}[F_{\tau^*+1:m}(t)])} \right).$$

**Remark 2.** Our scenario differs from the traditional nonparametric change point detection problem in a few ways: (i) Rather than testing the homogeneity of the original data sequence (which is the setup typically considered in the change point literature), we convert the string into a sequence of

p-values, based on which we conduct the change-point analysis; (ii) In the classical change point literature, the observations (in our case, the p-values) within the same segment are assumed to follow the same distribution. In contrast, for the watermark detection problem, the p-values from the watermarked segment could follow different distributions, adding a layer of difficulty to the analysis; (iii) The p-value sequence is dependent (where the strength of dependence is controlled by $B$), making our setup very different from the one in Carlstein [1988], which assumed the underlying data sequence to be independent; (iv) The technical tool used in our analysis must account for the particular dependence structure within the p-value sequence.

### 3.1 Binary segmentation

In this section, we describe an algorithm to separate watermarked and non-watermarked sub-strings by identifying multiple change point locations. There are two main types of algorithms for identifying multiple change points in the literature: (i) exact or approximate optimization by minimizing a penalized cost function [Harchaoui and Lévy-Leduc, 2010, Truong et al., 2020, R. Killick and Eckley, 2012, Li and Zhang, 2024, Zhang and Dawn, 2023] and (ii) approximate segmentation algorithms. Our proposed algorithm is based on the popular binary segmentation method, a top-down approximate segmentation approach for finding multiple change point locations. Initially proposed by Vostrikova [1981], binary segmentation identifies a single change point in a dataset using a CUSUM-like procedure and then employs a divide-and-conquer approach to find additional change points within sub-segments until a stopping condition is reached. However, as a greedy algorithm, binary segmentation can be less effective with multiple change points. Wild binary segmentation (WBS) [Fryzlewicz, 2014] and seeded binary segmentation (SeedBS) [Kovács et al., 2022] improve upon this by defining multiple segments to identify and aggregate potential change points. SeedBS additionally addresses the issue of overly long sub-segments in WBS that may contain several change points. SeedBS uses multiple layers of intervals, each with a fixed number of intervals of varying lengths and shifts, to enhance the search for change points. When comparing multiple candidates for the next change point, the narrowest-over-threshold (NOT) method [Baranowski et al., 2019] prioritizes narrower sub-segments. Built upon these ideas, we develop an effective algorithm to identify the change points that separate watermarked and non-watermarked sub-strings. The details are described in Algorithm 1 below.

---

**Algorithm 1** SeedBS-NOT for change point detection in potentially partially watermarked texts

---

**Require:** Sequence of p-values $\{p_i\}_{i=1}^m$, decay parameter $a \in [1/2, 1)$ for SeedBS, threshold $\zeta$ for NOT.
**Ensure:** Locations of the change points.
    Define $I_1 = (0, m]$.          ▷ Start of SeedBS
    **for** $k \leftarrow 2, \ldots, \lceil \log_{1/a} m \rceil$ **do**
        Number of intervals in the $k$-th layer: $n_k = 2\lceil (1/a)^{k-1} \rceil - 1$.
        Length of intervals in the $k$-th layer: $l_k = ma^{k-1}$.
        Shift of intervals in the $k$-th layer: $s_k = (m - l_k)/(n_k - 1)$.
        $k$-th layer intervals: $\mathcal{I}_k = \bigcup_{i=1}^{n_k} \{ (\lfloor (i-1)s_k \rfloor, \lceil (i-1)s_k + l_k \rceil] \}$.
    **end for**
    Define all seeded intervals $\mathcal{I} = \bigcup_{k=1}^{\lceil \log_{1/a} m \rceil} \mathcal{I}_k$.      ▷ End of SeedBS
    **for** $i \leftarrow 1, \ldots, |\mathcal{I}|$ **do**          ▷ Start of NOT
        Define the $i$-th interval $I_i = (r_i, s_i]$.
        Define $S_{r_i+1:s_i}(\tau) := \sup_{t \in [0,1]} \frac{(\tau - r_i)(s_i - \tau)}{(s_i - r_i)^{3/2}} |F_{r_i+1:\tau}(t) - F_{\tau+1:s_i}(t)|$.
        Let $\hat{\tau}_i = \arg\max_{r_i < \tau \leq s_i} S_{r_i+1:s_i}(\tau)$.
        Obtain $\tilde{p}_i$ through block bootstrap (10).
    **end for**
    Define the set of potential change point locations $\mathcal{O} = \{i : \tilde{p}_i < \zeta\}$ and the final set of change point locations $\mathcal{S} = \emptyset$.
    **while** $\mathcal{O} \neq \emptyset$ **do**
        Select $i = \arg\min_{i=1,\ldots,|\mathcal{O}|} \{|I_i|\} = \arg\min_{i=1,\ldots,|\mathcal{O}|} \{s_i - r_i\}$.
        $\mathcal{S} \leftarrow \mathcal{S} \cup \{\hat{\tau}_i\}$;     $\mathcal{O} \leftarrow \{j \leq |\mathcal{O}| : \hat{\tau}_i \notin I_j\}$.
    **end while**
    **return** $\mathcal{S}$.          ▷ End of NOT

---

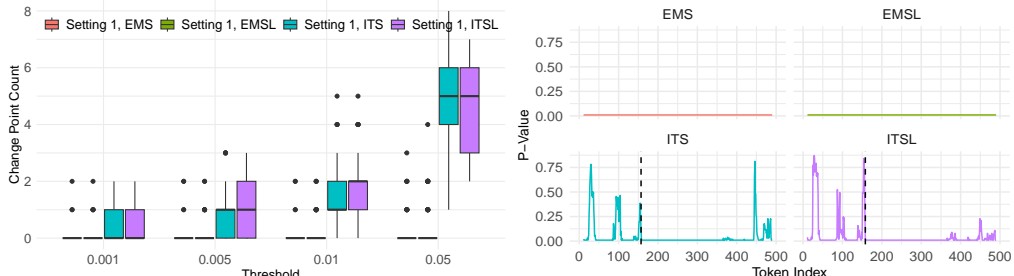

Figure 1: Left panel: boxplots of the number of false detections with respect to different thresholds $\zeta$. Right panel: sequences of $p$-values from different methods in Setting 1 for Prompt 1 with threshold $\zeta = 0.005$. The detected change point locations are marked with dashed lines at the index $157$ for `ITS` and $158$ for `ITSL`, respectively.

## 4 Numerical experiments

We conduct extensive real-data-based experiments following a similar empirical setting in Kirchenbauer et al. [2023a], where we generate watermarked text based on the prompts sampled from the news-like subset of the colossal clean crawled corpus (C4) dataset [Raffel et al., 2020]. We utilized three LLMs, namely `openai-community/gpt2` [Radford et al., 2019], `facebook/opt-1.3b` [Zhang et al., 2022] and `Meta-Llama-3-8B` [AI@Meta, 2024], to evaluate the effectiveness of the proposed method. We consider the following four watermark generation and detection methods:

- `ITS`: The inverse transform sampling method with the dependence measure defined in (1).
- `ITSL`: The inverse transform sampling method with the dependence measure defined in (B.2), which is based on the Levenshtein cost (B.1) with base alignment cost (B.3).
- `EMS`: The exponential minimum sampling method with the dependence measure defined in (2).
- `EMSL`: The exponential minimum sampling method with the dependence measure defined in (B.3), which is based on the Levenshtein cost (B.1) with base alignment cost (B.4).

The details of the Levenshtein cost are deferred to Appendix B.1. For each of the experiment settings, 100 prompts were used to generate the watermarked text. We fix the length of text $m = 500$, the size of sliding window $B = 20$, and the block size used in the block bootstrap-based test $B' = 20$. Results for other choices of $B$ are shown in Appendix C. In Algorithm 1, we set the decay parameter $a = \sqrt{2}$ and the minimum length of the intervals generated by SeedBS to be 50 such that the block bootstrap-based test is meaningful, and the threshold $\zeta \in \{0.05, 0.01, 0.005, 0.001\}$. We present the results for `openai-community/gpt2` in the main text and defer the results for `facebook/opt-1.3b` and `Meta-Llama-3-8B` to Appendix B.3 and Appendix B.4, respectively.

### 4.1 False positive analysis

We begin by analyzing the false discoveries of the change point detection method. We will generate watermarked text with a length of $m = 500$, where no change points exist.

- Setting 1 (no change point): Generate 500 tokens with a watermark.

The results for `openai-community/gpt2` are showed in Figure 1. The left panel illustrates that the two exponential minimum sampling methods result in fewer false discoveries compared to the two inverse transform sampling methods. Indeed, the existence of false discoveries highly depends on the quality of the obtained sequence of $p$-values. In the right panel, we fixed one prompt and plotted the sequence of $p$-values for the four methods. Most of the $p$-values are near 0. However, certain sub-strings have relatively high $p$-values for the two inverse transform methods, indicating that these methods failed to detect the watermark in these segments, resulting in false discoveries for change point detection.

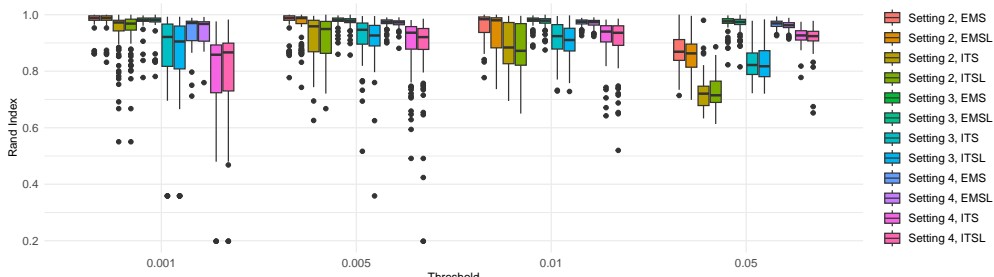

Figure 2: The boxplots of the Rand index comparing the clusters identified through the detected change points with the true clusters separated by the true change points with respect to different thresholds $\zeta$.

## 4.2 Change point analysis

When users modify the text generated by LLM, there may be some sub-strings with watermarks and others without. Our goal is to accurately separate the text into watermarked and non-watermarked sub-strings. In this section, we will focus on two types of attacks: insertion and substitution. To demonstrate, we will consider the following three settings:

- Setting 2 (insertion attack): Generate 250 tokens with watermarks, then append with 250 tokens without watermarks. In this setting, there is a single change point at the index 251.
- Setting 3 (substitution attack): Generate 500 tokens with watermarks, then substitute the token with indices ranging from 201 to 300 with non-watermarked text of length 100. In this setting, there are two change points at the indices 201 and 301, respectively.
- Setting 4 (insertion and substitution attacks): Generate 400 tokens with watermarks, substitute the token with indices ranging from 101 to 200 with non-watermarked text of length 100, and then insert 100 tokens without watermarks at the index 300. In this setting, there are four change points located at the indices 101, 201, 301, and 401, respectively.

For more complex simulation settings, please refer to Appendix B.5.

We compare the clusters identified through the detected change points with the true clusters separated by the true change points using the Rand index [Rand, 1971]. A higher Rand index indicates better performance of different methods.

Figure 2 shows the Rand index for four methods in Settings 2-4 corresponding to different thresholds $\zeta$. For each method, their performance in Settings 2-3 is better than that of Setting 4 when the threshold $\zeta \leq 0.01$. This is because Setting 4 includes two types of attacks, making the problem more difficult than in Settings 2 and 3. In all cases, the two exponential minimum sampling methods outperform the two inverse transform sampling methods, and EMS delivers the highest Rand index value. We want to emphasize again that the performance of the change point detection method highly depends on the quality of the obtained sequence of $p$-values. Figure 3 shows the $p$-value sequence for all methods given one fixed prompt in Setting 4. The change points detected by the EMS and EMSL methods are closer to the true change points compared to those detected by the ITS and ITSL methods. Additionally, the sequence of $p$-values for all methods in Settings 1-4 with the first 10 prompts extracted Google C4 dataset are shown in Figure B.1 in the Appendix.

## 5 Discussions

In this study, we have introduced a method for detecting whether a text is generated using LLM through randomization tests. We have demonstrated that our method effectively controls Type I and Type II errors under appropriate assumptions. Additionally, we have developed a technique to partition the published text into watermarked and non-watermarked sub-strings by treating it as a change point detection problem. Our proposed method accurately identifies watermarked sub-strings by determining the locations of change points. Simulation results using real data indicate that the EMS method outperforms the other methods.

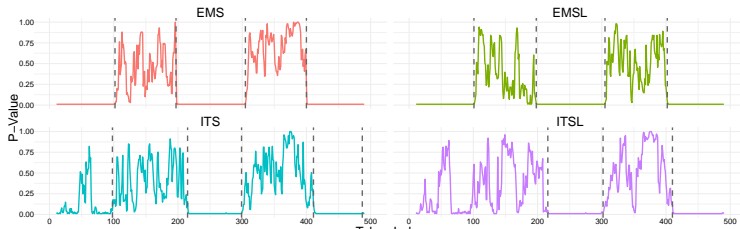

Figure 3: Sequences of $p$-values for all methods given one fixed prompt in Setting 4 with the threshold $\zeta = 0.005$. The true change points are located at 101, 201, 301, and 401. The change points detected by the EMS and EMSL methods are closer to the actual change points compared to those detected by the ITS and ITSL methods.

The performance of the segmentation algorithm depends crucially on the quality of the randomization-based $p$-values from each sub-string. Intuitively, a more significant discrepancy between the $p$-value distributions under the null and alternative will lead to better segmentation results. Thus, a powerful watermark detection algorithm is crucial to the success of the segmentation procedure. Motivated by Condition (4), an interesting future direction is to develop an adaptive watermark generation and detection procedure where the LLM provider adaptively embeds the key according to NTP, and the detector uses a corresponding adaptive procedure to detect the watermark.

Another interesting direction to explore is extending the algorithm to handle scenarios where the published text combines watermarked texts from different LLMs with varying watermark generation schemes. In this case, the goal is to separate the texts into sub-strings from different LLMs. This scenario involves multiple sequences of keys from different LLMs, each producing a sequence of $p$-values and change points. Figuring out how to aggregate these results to separate different LLMs and user-modified texts would be an intriguing problem.

## Acknowledgments and Disclosure of Funding

GL and XZ were supported by the National Institutes of Health under Grant R01GM144351. Part of the research was conducted using the Arseven Computing Cluster at the Department of Statistics, Texas A&M University.

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

# Appendix

## A  Proofs of the main results

*Proof of Theorem 1.*

(i) For simplicity of notation, denote $\varphi := \phi(\xi_{1:n}, \widetilde{y}_{1:m})$ and $\varphi^{(t)} := \phi(\xi_{1:n}^{(t)}, \widetilde{y}_{1:m})$ for $t = 1, \cdots, T$. Under the null hypothesis, we know that $\xi_{1:n}$ is independent of $\widetilde{y}_{1:m}$, which implies that the pairs $(\xi_{1:n}, \widetilde{y}_{1:m}), (\xi_{1:n}^{(1)}, \widetilde{y}_{1:m}), \ldots, (\xi_{1:n}^{(T)}, \widetilde{y}_{1:m})$ follow the same distribution. Hence $\varphi, \varphi^{(1)}, \ldots, \varphi^{(T)}$ are exchangeable. The exchangeability ensures that the rank of $\varphi$ relative to $\{\varphi, \varphi^{(1)}, \ldots, \varphi^{(t)}\}$ is uniformly distributed. Denote the order statistics $\varphi_{(1)} \leq \cdots \leq \varphi_{(T+1)}$. Then we have

$$P\left(\varphi = \varphi_{(j)}\right) = \frac{1}{T+1}, \qquad j = 1, \ldots, T+1.$$

Hence, for $j = 1, \ldots, T+1$, we have

$$P\left(p_T \leq \frac{j}{T+1}\right) = P\left(\varphi \in \{\varphi_{(T+2-j)}, \ldots, \varphi_{(T+1)}\}\right) = \frac{j}{T+1},$$

Then, we have $P\left(p_T \leq \alpha\right) = \lfloor(T+1)\alpha\rfloor/(T+1) \leq \alpha$.

(ii) Denote $\mathsf{E}_{\xi'} := \mathbb{E}[\phi(\xi'_{1:n}, \widetilde{y}_{1:m} | \mathcal{F}_m)]$. By Chebyshev's inequality and Condition (a), we get

$$P\left(|\phi(\xi'_{1:n}, \widetilde{y}_{1:m}) - \mathsf{E}_{\xi'}| \geq \epsilon/\sqrt{n} | \mathcal{F}_m\right) \leq \frac{\operatorname{Var}(\phi(\xi'_{1:n}, \widetilde{y}_{1:m}) | \mathcal{F}_m)}{\epsilon^2/n} \leq \frac{C_v}{\epsilon^2},$$

for all $\epsilon \geq 0$. Thus $\phi(\xi'_{1:n}, \widetilde{y}_{1:m}) - \mathsf{E}_{\xi'} = O_p(n^{-1/2})$, which together with Condition (b) implies that $\phi(\xi'_{1:n}, \widetilde{y}_{1:m}) = O_p(n^{-1/2})$ given $\mathcal{F}_m$.

Denote the distribution of $\phi(\xi'_{1:n}, \widetilde{y}_{1:m})$ conditional on $\mathcal{F}_m$ by $F$, and the empirical distribution of $\{\varphi^{(t)}\}_{t=0}^T$ by $F_T$, where we set $\varphi^{(0)} = \varphi$. Let $q_{1-\alpha, T} = \varphi_{(T+2-j_\alpha)}$ with $j_\alpha = \lfloor(T+1)\alpha\rfloor$. Note that $F_T(q_{1-\alpha, T}) = 1 - (j_\alpha - 1)/(T+1)$. Our test rejects the null whenever $p_T \leq \alpha$, which is equivalent to rejecting the null if $\varphi \geq \varphi_{(T+2-j_\alpha)}$. By the Dvoretzky-Kiefer-Wolfowitz inequality, we have

$$P(|F_T(q_{1-\alpha, T}) - F(q_{1-\alpha, T})| > \epsilon | \mathcal{F}_m) \leq P(\sup_x |F_T(x) - F(x)| > \epsilon | \mathcal{F}_m) \leq C_1 \exp(-2T\epsilon^2)$$

for some constant $C_1 > 0$, which implies that with probability greater than $1 - C_1 \exp(-2T\epsilon^2)$, $F(q_{1-\alpha, T}) < 1 - (j_\alpha - 1)/(T+1) + 2\epsilon$. Define $F^{-1}(t) = \inf\{s : F(s) \geq t\}$ and the event $\mathcal{A}_T = \{q_{1-\alpha, T} < F^{-1}(1 - (j_\alpha - 1)/(T+1) + 2\epsilon)\}$. Then we have $P(\mathcal{A}_T | \mathcal{F}_m) \geq 1 - C_1 \exp(-2T\epsilon^2)$. In addition, as $\phi(\xi'_{1:n}, \widetilde{y}_{1:m}) = O_p(n^{-1/2})$, we have $F^{-1}(s) = O(n^{-1/2})$ for any $s < 1$. By Condition (a), we have $\sqrt{n}(\phi(\xi_{1:n}, \widetilde{y}_{1:m}) - \mathbb{E}[\phi(\xi_{1:n}, \widetilde{y}_{1:m}) | \mathcal{F}_m]) = O_p(1)$. Hence, for $T > 2/\epsilon - 1$,

$$P(\phi(\xi_{1:n}, \widetilde{y}_{1:m}) \geq q_{1-\alpha, T} | \mathcal{F}_m)$$
$$\geq P\left(\sqrt{n}(\phi(\xi_{1:n}, \widetilde{y}_{1:m}) - \mathbb{E}[\phi(\xi_{1:n}, \widetilde{y}_{1:m}) | \mathcal{F}_m]) + \sqrt{n}\mathbb{E}[\phi(\xi_{1:n}, \widetilde{y}_{1:m}) | \mathcal{F}_m] \geq \sqrt{n}q_{1-\alpha, T}, \mathcal{A}_T | \mathcal{F}_m\right)$$
$$\geq P\Big(\sqrt{n}(\phi(\xi_{1:n}, \widetilde{y}_{1:m}) - \mathbb{E}[\phi(\xi_{1:n}, \widetilde{y}_{1:m}) | \mathcal{F}_m]) + \sqrt{n}\mathbb{E}[\phi(\xi_{1:n}, \widetilde{y}_{1:m}) | \mathcal{F}_m]$$
$$\geq \sqrt{n}F^{-1}(1 - (j_\alpha - 1)/(T+1) + 2\epsilon), \mathcal{A}_T | \mathcal{F}_m\Big)$$
$$\geq P\Big(\sqrt{n}(\phi(\xi_{1:n}, \widetilde{y}_{1:m}) - \mathbb{E}[\phi(\xi_{1:n}, \widetilde{y}_{1:m}) | \mathcal{F}_m]) + \sqrt{n}\mathbb{E}[\phi(\xi_{1:n}, \widetilde{y}_{1:m}) | \mathcal{F}_m]$$
$$\geq \sqrt{n}F^{-1}(1 - \alpha + 3\epsilon), \mathcal{A}_T | \mathcal{F}_m\Big)$$
$$\geq 1 - C_1 \exp(-2T\epsilon^2) + o(1),$$

where we have used Condition (c) and the fact that $\sqrt{n}F^{-1}(1 - \alpha + 3\epsilon) = O(1)$ to get the convergence. □

We state the following Lemma, which is useful for the proof of Corollary 1.

**Lemma 1.** $\{\xi_i\}_{i=1}^n$ are conditionally independent given $y_{-n_0:n}$.

*Proof of Lemma 1.* Recall that $y_i = \Gamma(\xi_i, p(\cdot|y_{-n_0:i-1})) = \Gamma_i(\xi_i)$, where $\Gamma_i$ depends on $y_{-n_0:i-1}$. We note that

$$p(\xi_{1:n}, y_{1:n}|y_{-n_0:0}) = p(\xi_1, y_1|y_{-n_0:0}) \prod_{i=2}^{n} p(\xi_i, y_i|\xi_{1:i-1}, y_{-n_0:i-1}) = \prod_{i=1}^{n} \mathbf{1}\{\Gamma_i(\xi_i) = y_i\} p(\xi_i).$$

Hence the conditional distribution of $\xi_{1:n}$ given $y_{-n_0:n}$ is equal to

$$p(\xi_{1:n}|y_{-n_0:n}) = \prod_{i=1}^{n} \frac{\mathbf{1}\{\Gamma_i(u) = y_i\} d\nu(u)}{\int_{\Gamma_i(u)=y_i} d\nu(u)}, \tag{A.1}$$

which implies that $\{\xi_i\}_{i=1}^{n}$ are conditionally independent given $y_{-n_0:n}$. $\qquad\square$

*Proof of Corollary 1.*
(i) Recall in Example 1 that the test statistic is given by:

$$\phi(\xi_{1:n}, y_{1:n}) = \frac{1}{n} \sum_{i=1}^{n} (u_i - 1/2) \left( \frac{\pi_i(y_i) - 1}{V - 1} - \frac{1}{2} \right) := \frac{1}{n} \sum_{i=1}^{n} h_i(\xi_i, y_i),$$

where $\xi_i = (u_i, \pi_i)$. We first note that $h_i(\xi_i, y_i)$ is bounded and thus Condition (a) of Theorem 1 holds. Since $\xi'_{1:n}$ is independent of $y_{-n_0:n}$, we have $u'_i|y_{1:n} \sim \text{Unif}[0, 1]$. Thus, $\mathbb{E}[\phi(\xi'_{1:n}, y_{1:n})|y_{1:n}] = 0$, which implies that Condition (b) of Theorem 1 is fulfilled.

Denote $\mu_i(\cdot) = p(\cdot|y_{-n_0:i-1})$. Conditional on $y_{-n_0:i}$ and $\pi_i(y_i)$, we know that $u_i$ follows the uniform distribution over the interval $[\mu_i(y : \pi_i(y) < \pi_i(y_i)), \mu_i(y : \pi_i(y) \le \pi_i(y_i))]$. As a result, we can calculate the expected value of $u_i$ given $y_{-n_0:i}$ and $\pi_i(y_i)$ as

$$\begin{aligned}
\mathbb{E}[u_i|y_{-n_0:i}, \pi_i(y_i)] &= \frac{1}{2} \{\mu_i(y : \pi_i(y) < \pi_i(y_i)) + \mu_i(y : \pi_i(y) \le \pi_i(y_i))\} \\
&= \frac{\mu_i(y_i)}{2} + \mu_i(y : \pi_i(y) < \pi_i(y_i)) \\
&= \frac{\mu_i(y_i)}{2} + \frac{\pi_i(y_i) - 1}{V - 1} (1 - \mu_i(y_i)) \\
&= \frac{1}{2} + (1 - \mu_i(y_i)) \left( \frac{\pi_i(y_i) - 1}{V - 1} - \frac{1}{2} \right),
\end{aligned}$$

where the third equality is because given $\pi_i(y_i) = k$, $\pi_i$ follows the uniform distribution over the permutation space with the restriction $\pi_i(y_i) = k$. Then, we have

$$\begin{aligned}
&\mathbb{E}\left[ (u_i - 1/2) \left( \frac{\pi_i(y_i) - 1}{V - 1} - \frac{1}{2} \right) | y_{-n_0:i} \right] \\
={}& \mathbb{E}\left[ \mathbb{E}\left[ (u_i - 1/2) \left( \frac{\pi_i(y_i) - 1}{V - 1} - \frac{1}{2} \right) | y_{-n_0:i}, \pi_i(y_i) \right] | y_{-n_0:i} \right] \\
={}& \mathbb{E}\left[ (1 - \mu_i(y_i)) \left( \frac{\pi_i(y_i) - 1}{V - 1} - \frac{1}{2} \right)^2 \right] \\
={}& (1 - p(y_i|y_{-n_0:i-1})) \mathbb{E}\left[ \left( \frac{\pi_i(y_i) - 1}{V - 1} - \frac{1}{2} \right)^2 | y_{-n_0:i} \right].
\end{aligned}$$

Given $y_i$, $(\pi_i(y_i) - 1)/(V - 1)$ follows the uniform distribution over the discrete space $\{0, 1/(V - 1), \ldots, 1\}$. Thus we have $\mathbb{E}[(\pi_i(y_i) - 1)/(V - 1)] = 1/2$ and $\text{Var}((\pi_i(y_i) - 1)/(V - 1)) = C$, which is a constant less than $1/12$ (i.e., the variance of $\text{Unif}[0, 1]$). Hence, $\mathbb{E}[\phi(\xi_{1:n}, y_{1:n})] = Cn^{-1} \sum_{i=1}^{n} (1 - p(y_i|y_{1:i-1}))$ and Condition (c) of Theorem 1 is satisfied due to (4).

(ii) In Example 2, the test statistic is given by

$$\phi(\xi_{1:n}, y_{1:n}) = \frac{1}{n} \sum_{i=1}^{n} \{\log(\xi_{i,y_i}) + 1\}.$$

Observe that $E_{ik} := -\log(\xi_{ik})/p(k|y_{-n_0:i-1}) \sim \text{Exp}(p(k|y_{-n_0:i-1}))$. Since $\xi'_{1:n}$ is independent of $y_{1:n}$, we have $-\log(\xi'_{i,y_i})|y_{-n_0:n} \sim \text{Exp}(1)$. Hence, conditional on $y_{-n_0:n}$, we have

$$\mathbb{E}[\phi(\xi'_{1:n}, y_{1:n})|y_{-n_0:n}] = 0, \quad \text{Var}(\phi(\xi'_{1:n}, y_{1:n})|y_{-n_0:n}) = \frac{1}{n}.$$

Condition (b) of Theorem 1 is satisfied.

Given $y_{-n_0:n}$, we know $E_{i,y_i} = \min_{1 \le k \le V} E_{ik}$, which implies $-\log(\xi_{i,y_i})/p(y_i|y_{-n_0:i-1})|y_{-n_0:n} \sim \text{Exp}(1)$. It is worth noting that

$$P(-\log(\xi_{i,y_i}) \ge t) = p\left(-\frac{\log(\xi_{i,y_i})}{p(y_i|y_{-n_0:i-1})} \ge \frac{t}{p(y_i|y_{-n_0:i-1})}\right) = \exp\left(-\frac{t}{p(y_i|y_{-n_0:i-1})}\right).$$

That is $-\log(\xi_{i,y_i})|y_{-n_0:n} \sim \text{Exp}(1/p(y_i|y_{-n_0:i-1}))$. According to Lemma 1, $\xi_i$s are conditionally independent given $y_{-n_0:n}$. Thus, we have

$$\mathbb{E}[\phi(\xi_{1:n}, y_{1:n})|y_{-n_0:n}] = \frac{1}{n}\sum_{i=1}^{n}(1 - p(y_i|y_{-n_0:i-1})),$$

$$\text{Var}(\phi(\xi_{1:n}, y_{1:n})|y_{-n_0:n}) = \frac{1}{n^2}\sum_{i=1}^{n}p(y_i|y_{-n_0:i-1})^2 \le \frac{1}{n}.$$

Therefore, Conditions (a) and (c) of Theorem 1 are satisfied. $\qquad\square$

*Proof of Theorem 2.* We only prove the result under the setting of Example 1 as the proof for Example 2 is similar with the help of Bernstein's inequality. Because $|h_i| \le 1/4$, by Hoeffding's inequality, we have

$$P\left(|\mathcal{M}(\xi_{a:a+B-1}, \widetilde{y}_{b:b+B-1}) - \mathbb{E}[\mathcal{M}(\xi_{a:a+B-1}, \widetilde{y}_{b:b+B-1})|\widetilde{y}_{1:m}, y_{-n_0:n}]| > t|\widetilde{y}_{1:m}, y_{-n_0:n}\right)$$
$$\le 2\exp\left(-8Bt^2\right).$$

By the union bound, we have

$$P\left(\max_{1 \le a \le n-B+1, 1 \le b \le m-B+1} |\mathcal{M}(\xi_{a:a+B-1}, \widetilde{y}_{b:b+B-1})\right.$$

$$\left. - \mathbb{E}[\mathcal{M}(\xi_{a:a+B-1}, \widetilde{y}_{b:b+B-1})|\widetilde{y}_{1:m}, y_{-n_0:n}]| > t|\widetilde{y}_{1:m}, y_{-n_0:n}\right)$$

$$\le 2(n-B+1)(m-B+1)\exp\left(-8Bt^2\right).$$

Integrating out the strings in $y_{1:n}$ that are not contained in $\widetilde{y}_{1:m}$, we obtain

$$P\left(\max_{1 \le a \le n-B+1, 1 \le b \le m-B+1} |\mathcal{M}(\xi_{a:a+B-1}, \widetilde{y}_{b:b+B-1})\right.$$

$$\left. - \mathbb{E}[\mathcal{M}(\xi_{a:a+B-1}, \widetilde{y}_{b:b+B-1})|\widetilde{y}_{1:m}, y_{-n_0:n}]| > t|\mathcal{F}_m\right)$$

$$\le 2(n-B+1)(m-B+1)\exp\left(-8Bt^2\right),$$

where $\mathcal{F}_m = [\widetilde{y}_{1:m}, y_{-n_0:0}]$. Thus, conditional on $\mathcal{F}_m$, $\max_{a,b}\{|\mathbb{E}[\mathcal{M}(\xi_{a:a+B-1}, \widetilde{y}_{b:b+B-1})|\mathcal{F}_m] - \mathcal{M}(\xi_{a:a+B-1}, \widetilde{y}_{b:b+B-1})|\} = O(C_{N,B})$. Note that

$$\phi(\xi_{1:n}, \widetilde{y}_{1:m}) = \max_{a,b} \mathcal{M}(\xi_{a:a+B-1}, \widetilde{y}_{b:b+B-1})$$

$$\ge \max_{a,b} \mathbb{E}[\mathcal{M}(\xi_{a:a+B-1}, \widetilde{y}_{b:b+B-1})|\mathcal{F}_m]$$

$$- \max_{a,b}\{\mathbb{E}[\mathcal{M}(\xi_{a:a+B-1}, \widetilde{y}_{b:b+B-1})|\mathcal{F}_m] - \mathcal{M}(\xi_{a:a+B-1}, \widetilde{y}_{b:b+B-1})\}$$

$$= \max_{a,b} \mathbb{E}[\mathcal{M}(\xi_{a:a+B-1}, \widetilde{y}_{b:b+B-1})|\mathcal{F}_m] + O(C_{N,B}).$$

On the other hand, for a randomly generated key $\xi'_{1:n}$, $\mathbb{E}[\mathcal{M}(\xi'_{a:a+B-1}, \widetilde{y}_{b:b+B-1})|\mathcal{F}_m] = 0$ for all $a, b$. By the same argument, we get

$$P\left(\phi(\xi'_{1:n}, \widetilde{y}_{1:m}) > t|\mathcal{F}_m\right) = P\left(\max_{1 \leq a \leq n-B+1, 1 \leq b \leq m-B+1} \mathcal{M}(\xi'_{a:a+B-1}, \widetilde{y}_{b:b+B-1}) > t\Big|\mathcal{F}_m\right)$$
$$\leq 2(n-B+1)(m-B+1)\exp\left(-8Bt^2\right),$$

which suggests that $F^{-1}(s) = O(C_{N,B})$ with $F$ being the distribution of $\phi(\xi'_{1:n}, \widetilde{y}_{1:m})$ conditional on $\mathcal{F}_m$ and $F^{-1}(t) = \inf\{s : F(s) \geq t\}$. The rest of the arguments are similar to those in the proof of Theorem 1. We skip the details. $\qquad\square$

*Proof of Proposition 1.* Note that conditional on $\mathcal{F}_m$, the $p$-value sequence is $B$-dependent in the sense that $p_i$ and $p_j$ are independent only if $|i - j| > B$. Under the assumption on $B$, we have

$$\text{Var}(F_{a+1:b}(t)|\mathcal{F}_m) = O\left(\frac{B}{|b-a|}\right) = o(1)$$

for $|b - a| \asymp m$, which implies that $F_{a+1:b}(t) - \mathbb{E}[F_{a+1:b}(t)] \to^p 0$ for any given $t \in [0, 1]$. Using similar arguments as in the proof of Theorem 7.5.2 of Resnick [2019], we can strengthen the result to allow uniform convergence over $t \in [0, 1]$:

$$\sup_{t \in [0,1]} |F_{a+1:b}(t) - \mathbb{E}[F_{a+1:b}(t)]| \to^p 0.$$

Therefore, we obtain

$$\max_{1 \leq \tau < m} S_{1:m}(\tau)$$
$$\geq S_{1:m}(\tau^*)$$
$$= \sqrt{m}\frac{\tau^*(m-\tau^*)}{m^2}\sup_{t \in [0,1]}|F_{1:\tau^*}(t) - t - (F_{\tau^*+1:m}(t) - \mathbb{E}[F_{\tau^*+1:m}(t)]) + t - \mathbb{E}[F_{\tau^*+1:m}(t)]|$$
$$\geq \sqrt{m}\frac{\tau^*(m-\tau^*)}{m^2}\left\{\sup_{t \in [0,1]}|t - \mathbb{E}[F_{\tau^*+1:m}(t)]| - \sup_{t \in [0,1]}|F_{1:\tau^*}(t) - t|\right.$$
$$\left. - \sup_{t \in [0,1]}|F_{\tau^*+1:m}(t) - \mathbb{E}[F_{\tau^*+1:m}(t)]|\right\}$$
$$= \sqrt{m}\gamma^*(1 - \gamma^*)\{D(F_0, \mathbb{E}[F_{\tau^*+1:m}(t)]) + o_p(1)\}(1 + o(1)) \to +\infty.$$
$$\qquad\square$$

*Proof of Theorem 3.* We begin the proof by noting that

$$\frac{\tau(m-\tau)}{m^{3/2}}(F_{1:\tau}(t) - F_{\tau+1:m}(t)) = \frac{1}{\sqrt{m}}\sum_{i=1}^{\tau}(\mathbf{1}\{p_i \leq t\} - F_{1:m}(t)).$$

Let us focus on the case where $\hat{\tau} \leq \tau^*$. The other case where $\hat{\tau} > \tau^*$ can be proved in a similar way. By the definition of $\hat{\tau}$, we have

$$0 \leq \frac{1}{m}\sup_t\left|\sum_{i=1}^{\hat{\tau}}(\mathbf{1}\{p_i \leq t\} - F_{1:m}(t))\right| - \frac{1}{m}\sup_t\left|\sum_{i=1}^{\tau^*}(\mathbf{1}\{p_i \leq t\} - F_{1:m}(t))\right| := I(\hat{\tau}) - I(\tau^*).$$

For any $1 \leq a < b \leq m$ and there is no change point between $[a, b]$, define $W_{a:b}(t) = \sum_{i=a}^{b}\mathbf{1}\{p_i \leq t\}$ and $\bar{W}_{a:b}(t) = \sum_{i=a}^{b}\{\mathbf{1}\{p_i \leq t\} - \mathbb{E}[F_{a:b}(t)]\}$. For any $\tau \leq \tau^*$, we have

$$I(\tau) = \frac{1}{m}\sup_t\left|W_{1:\tau}(t) - \frac{\tau}{m}(W_{1:\tau^*}(t) + W_{\tau^*+1:m}(t))\right|$$
$$= \frac{1}{m}\sup_t\left|\tau F_0(t) - \frac{\tau}{m}\{\tau^* F_0(t) + (m - \tau^*)\mathbb{E}[F_{\tau^*+1:m}(t)]\}\right| + R(\tau)$$
$$= \frac{(m-\tau^*)\tau}{m^2}D(F_0, \mathbb{E}[F_{\tau^*+1:m}(t)]) + R(\tau),$$

where $R(\tau)$ is a reminder term satisfying that

$$|R(\tau)| \leq \frac{1}{m} \sup_t \left| \bar{W}_{1:\tau}(t) - \frac{\tau}{m} \left( \bar{W}_{1:\tau^*}(t) + \bar{W}_{\tau^*+1:m}(t) \right) \right|.$$

Hence, we have

$$
\begin{aligned}
0 \leq & I(\hat{\tau}) - I(\tau^*) \\
= & \frac{(m - \tau^*)(\hat{\tau} - \tau^*)}{m^2} D(F_0, \mathbb{E}[F_{\tau^*+1:m}(t)]) + R(\hat{\tau}) - R(\tau^*) \\
\leq & \frac{(m - \tau^*)(\hat{\tau} - \tau^*)}{m^2} D(F_0, \mathbb{E}[F_{\tau^*+1:m}(t)]) + 2 \sup_{\tau \leq \tau^*} |R(\tau)|,
\end{aligned}
$$

which implies that

$$\tau^* - \hat{\tau} \leq \frac{2m^2}{(m - \tau^*) D(F_0, \mathbb{E}[F_{\tau^*+1:m}(t)])} \sup_{\tau \leq \tau^*} |R(\tau)|.$$

Notice that

$$
\begin{aligned}
\sup_{\tau} |R(\tau)| \leq & \frac{1}{m} \sup_{\tau \leq \tau^*} \sup_{t \in [0,1]} \left| \bar{W}_{1:\tau}(t) - \frac{\tau}{m} \{ \bar{W}_{1:\tau^*}(t) + \bar{W}_{\tau^*+1:m}(t) \} \right| \\
\leq & \frac{1}{m} \sup_{\tau \leq \tau^*} \sup_{t \in [0,1]} \left| \bar{W}_{1:\tau}(t) \right| + \frac{1}{m} \sup_{t \in [0,1]} \left| \bar{W}_{1:\tau^*}(t) + \bar{W}_{\tau^*+1:m}(t) \right| \\
\leq & \frac{1}{m} \sup_{\tau \leq m} \sup_{t \in [0,1]} \left| \bar{W}_{1:\tau}(t) \right| + O_p(m^{-1/2}).
\end{aligned}
$$

It remains to analyze $\sup_{\tau \leq m} \sup_{t \in [0,1]} \left| \bar{W}_{1:\tau}(t) \right|$. We first present a result that slightly modifies the Ottaviani's inequality. For the ease of notation, we write $\| \bar{W}_{1:\tau} \| = \sup_{t \in [0,1]} \left| \bar{W}_{1:\tau}(t) \right|$. For any $u > 0$ and $v > B$, we have

$$P \left( \max_{\tau \leq m} \| \bar{W}_{1:\tau} \| > u + v \right) \leq \frac{P(\| \bar{W}_{1:m} \| > v - B)}{1 - \max_\tau P(\| \bar{W}_{\tau+1:m} \| > u)}. \tag{A.2}$$

To see this, let $A_k$ be the event that $\| \bar{W}_{1:k} \|$ is the first $\| \bar{W}_{1:j} \|$ (for $j = 1, 2, \ldots, m$) that is strictly greater than $u + v$. The event on the LHS is the disjoint union of $A_1, \ldots, A_m$. Note that $\| \bar{W}_{k+1+B:m} \|$ is independent of $\| \bar{W}_1 \|, \ldots, \| \bar{W}_k \|$ (and hence is independent of $A_k$). We have

$$
\begin{aligned}
P(A_k) \min_{0 \leq \tau < m} P(\| \bar{W}_{\tau+1:m} \| \leq u) \leq & P(A_k, \| \bar{W}_{k+1+B:m} \| \leq u) \\
\leq & P(A_k, \| \bar{W}_{k+1:m} \| \leq u + B) \\
\leq & P(A_k, \| \bar{W}_{1:m} \| > v - B),
\end{aligned}
$$

where we have usede the fact $\| \bar{W}_{k+1:m} \| \leq \| \bar{W}_{k+1:k+B} \| + \| \bar{W}_{k+B+1:m} \| \leq u + B$. Summing over $k$ leads to

$$P \left( \max_{\tau \leq m} \| \bar{W}_{1:\tau} \| > u + v \right) \min_{0 \leq \tau < m} P(\| \bar{W}_{\tau+1:m} \| \leq u) \leq P(\| \bar{W}_{1:m} \| > v - B),$$

which gives the desired result.

Next, we study $P(\| \bar{W}_{\tau+1:m} \| > u)$ for any $\tau = 0, \ldots, m - 1$. Note that $P(\| \bar{W}_{\tau+1:m} \| > u) \leq P(\| \bar{W}_{\tau+1:\tau^*} \| > u/2) + P(\| \bar{W}_{\tau^*+1:m} \| > u/2)$. Thus, without loss of generality, let us focus our analysis on the probability $P(\| \bar{W}_{a+1:b} \| > u)$, where the corresponding segment does not contain a change point. Assume $b - a = 2KB$. (Notice that for general $a < b$, we have an additional interval whose length is smaller than $B$. Hence, a similar analysis below can be used.) We divide the index set $\{a + 1, a + 2, \ldots, b\}$ into $2K$ consecutive blocks, denoted by $J_1, \ldots, J_{2K}$, with equal block size $B$. Then we have

$$P(\| \bar{W}_{a+1:b} \| > u) \leq P \left( \left\| \sum_{i=1}^K \bar{F}_{J_{2i-1}} \right\| > u/(2B) \right) + P \left( \left\| \sum_{i=1}^K \bar{F}_{J_{2i}} \right\| > u/(2B) \right)$$

with $\bar{F}_{J_i} = \bar{W}_{J_i}/B$, where $\sum_{i=1}^{K} \bar{F}_{J_{2i-1}}$ and $\sum_{i=1}^{K} \bar{F}_{J_{2i}}$ are both sums of independent bounded random variables. Let us analyze the second term on the RHS. Define

$$G(t) = \sum_{i=1}^{K} \sum_{j \in J_{2i}} P(p_j \leq t)/(KB),$$

$$G_m(t) = \sum_{i=1}^{K} \sum_{j \in J_{2i}} \mathbf{1}\{p_j \leq t\}/(KB).$$

Let $t_{v,L} = G^{-1}(v/L)$ for $v = 1, 2, \ldots, L$. Following the argument in Theorem 7.5.2 of Resnick [2019], we can show that

$$\frac{1}{K}\left\|\sum_{i=1}^{K} \bar{F}_{J_{2i}}\right\| = \|G_m - G\| \leq \max_{1 \leq v \leq L} |G_m(t_{v,L}) - G(t_{v,L})| \vee |G_m(t_{v,L}-) - G(t_{v,L}-)| + \frac{1}{L},$$

where $G_m(t-) = \sum_{i=1}^{K} \sum_{i \in J_{2i}} \mathbf{1}\{p_i < t\}/(KB)$ and $G(t-)$ is defined similarly. Thus we have

$$P\left(\left\|\sum_{i=1}^{K} \bar{F}_{J_{2i}}\right\| > u/(2B)\right)$$

$$\leq P\left(\max_{1 \leq v \leq L} |G_m(t_{v,L}) - G(t_{v,L})| \vee |G_m(t_{v,L}-) - G(t_{v,L}-)| + 1/L > u/(b-a)\right)$$

$$\leq \sum_{v=1}^{L} P\left(|G_m(t_{v,L}) - G(t_{v,L})| \vee |G_m(t_{v,L}-) - G(t_{v,L}-)| > u/(b-a) - 1/L\right)$$

$$\leq C_1 L \exp\left(-C_2 K(u/(b-a) - 1/L)^2\right)$$

where the third inequality follows from Hoeffding's inequality. For any $\epsilon > 0$, we can set $u = C_3(b-a)\sqrt{\log(K)}/\sqrt{K} = 2C_3 B\sqrt{K \log(K)}$ and $L = \sqrt{K}$ for some large enough $C_3$ such that

$$P\left(\left\|\sum_{i=1}^{K} \bar{F}_{J_{2i}}\right\| > u/(2B)\right) \leq \epsilon.$$

Now back to (A.2), set $u = C_4\sqrt{mB \log(m/B)}$ and $v = C_5\sqrt{mB \log(m/B)}$. We can make the RHS of (A.2) arbitrarily small with large enough $C_4$ and $C_5$. It thus gives $\max_{\tau \leq m} \|\bar{S}_{1:\tau}\| = O(\sqrt{mB \log(m/B)})$ and $\sup_\tau |R(\tau)| = O\left(\sqrt{B \log(m/B)/m}\right)$. Hence, we deduce that

$$\tau^* - \hat{\tau} \leq O_p\left(\frac{\sqrt{mB \log(m/B)}}{D(F_0, \mathbb{E}[F_{\tau^*+1:m}(t)])}\right).$$

A similar argument applies to the other direction, which gives

$$|\hat{\tau} - \tau^*| = O_p\left(\frac{\sqrt{mB \log(m/B)}}{D(F_0, \mathbb{E}[F_{\tau^*+1:m}(t)])}\right).$$

$\square$

# B   Additional numerical results

## B.1   The Levenshtein cost

Recall that we use $\mathcal{V}$ to denote the vocabulary and $\Xi$ to represent the space of watermark keys. Let $\mathcal{V}^*$ be the space of strings, where $*$ can be any positive integer; for example, $\widetilde{y}_{1:m} \in \mathcal{V}^m$. Similarly, we define $\Xi^*$ as the space of watermark key sequences. Given a string $\widetilde{y}$, let $\widetilde{y}_{a:}$ be the string from the $a$th token to the end; for example, if $\widetilde{y} = \widetilde{y}_{1:m}$, then $\widetilde{y}_{2:} = \widetilde{y}_{2:m}$. Denote the length of a string $\widetilde{y}$ as $\mathtt{len}(\widetilde{y})$.

**Definition 1** (Simple Levenshtein cost). *Let $\gamma \in \mathbb{R}$ and base alignment cost $d_0 : \mathcal{V} \times \Xi \to \mathbb{R}$. Given a string $\widetilde{y} \in \mathcal{V}^*$ and a watermark key sequence $\xi \in \Xi^*$, the simple Levenshtein cost $d_\gamma(\widetilde{y}, \xi)$ is defined by*

$$d_\gamma(\widetilde{y}, \xi) := \min\left(d_\gamma(\widetilde{y}_{2:}, \xi_{2:}) + d_0(y_1, \xi_1), d_\gamma(\widetilde{y}, \xi_{2:}) + \gamma, d_\gamma(\widetilde{y}_{2:}, \xi) + \gamma\right), \tag{B.1}$$

*with $d_\gamma(\widetilde{y}, \xi) := \gamma \cdot \mathtt{len}(\widetilde{y})$ if $\xi$ is empty and $d_\gamma(\widetilde{y}, \xi) := \gamma \cdot \mathtt{len}(\xi)$ if $\widetilde{y}$ is empty.*

We use the following metric to quantify the dependence between the watermark key and the string:

$$\mathcal{M}(\xi_{1:n}, \widetilde{y}_{1:m}) = -d_\gamma(\widetilde{y}_{1:m}, \xi_{1:n}), \tag{B.2}$$

where $\gamma = 0.4$, and

$$d_0(\widetilde{y}_1, (u_1, \pi_1)) = \left| u_1 - \frac{\pi_1(\widetilde{y}_1) - 1}{|\mathcal{V}| - 1} \right|, \tag{B.3}$$

in inverse transform sampling method, while

$$d_0(\widetilde{y}_1, \xi_1) = \log\left(1 - \xi_{1, \widetilde{y}_1}\right), \tag{B.4}$$

in exponential minimum sampling method.

## B.2   Sequences of $p$-values from OpenAI-Community/GPT2

Figure B.1 presents the $p$-values for $500$ text tokens generated from $10$ prompts. The language model used is `openai-community/gpt2` obtained from `https://huggingface.co/openai-community/gpt2`. The $p$-value sequences are organized into four groups, namely four settings corresponding to the numerical experiments section in the main paper (Section 4):

- Setting 1 (no change point): Generate 500 tokens with a watermark.
- Setting 2 (insertion attack): Generate 250 tokens with watermarks, then append with 250 tokens without watermarks. In this setting, there is a single change point at the index 251.
- Setting 3 (substitution attack): Generate 500 tokens with watermarks, then substitute the token with indices ranging from 201 to 300 with non-watermarked text of length 100. In this setting, there are two change points at the indices 201 and 301.
- Setting 4 (insertion and substitution attacks): Generate 400 tokens with watermarks, substitute the token with indices ranging from 101 to 200 with non-watermarked text of length 100, and then insert 100 tokens without watermarks at the index 300. In this setting, there are four change points located at the indices 101, 201, 301, and 401.

Within each group, the rows represent $p$-values calculated using four different distance metrics: `EMS`, `EMSL`, `ITS`, and `ITSL`. It is easy to see that `EMS` performs the best, followed by `EMSL`.

## B.3   Results for Facebook/OPT-1.3b

Figure B.2 shows the results for false discoveries under Setting 1, where the text tokens are all watermarked and generated with `facebook/opt-1.3b`. A lower threshold leads to fewer false discoveries.

Figure B.4 shows the boxplots of the Rand index for the four methods under different settings. `EMS` demonstrates the best performance across all settings, achieving its best performance at a threshold of $\zeta = 0.005$.

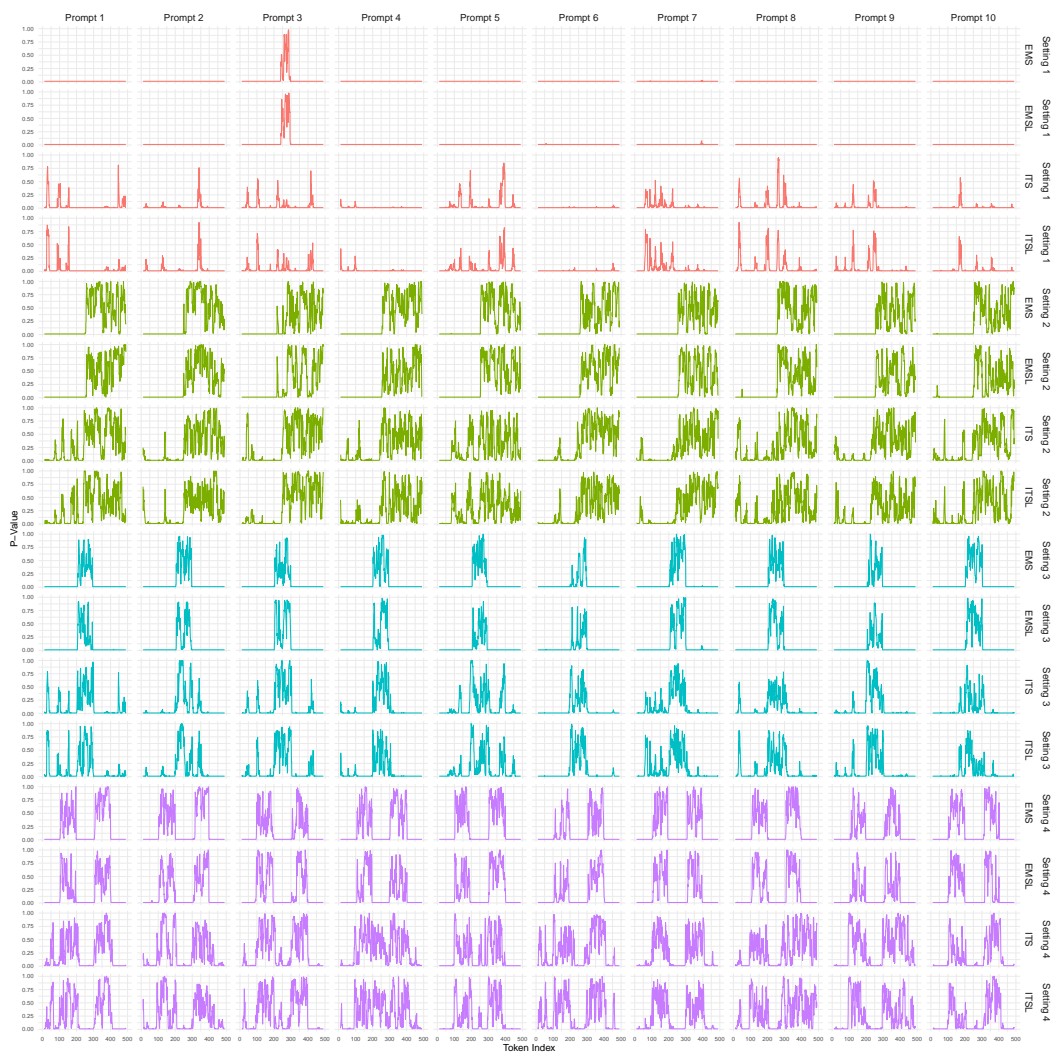

Figure B.1: Sequences of $p$-values for the first 10 prompts extracted from the Google C4 dataset for LLM `openai-community/gpt2`, organized into groups of four consecutive rows, each group corresponding to a distinct setting. Within each group, the rows represent $p$-values calculated using four different distance metrics: EMS, EMSL, ITS, and ITSL.

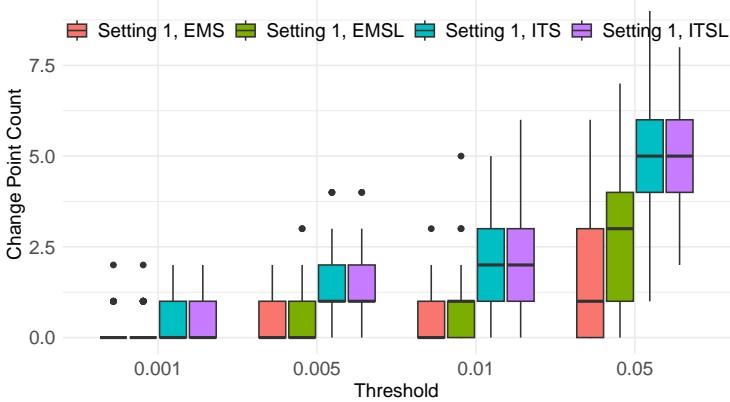

Figure B.2: The boxplots of the number of false positives with respect to different thresholds $\zeta$ under Setting 1. The texts are generated using `facebook/opt-1.3b`, and the four distance metrics are EMS, EMSL, ITS, and ITSL.

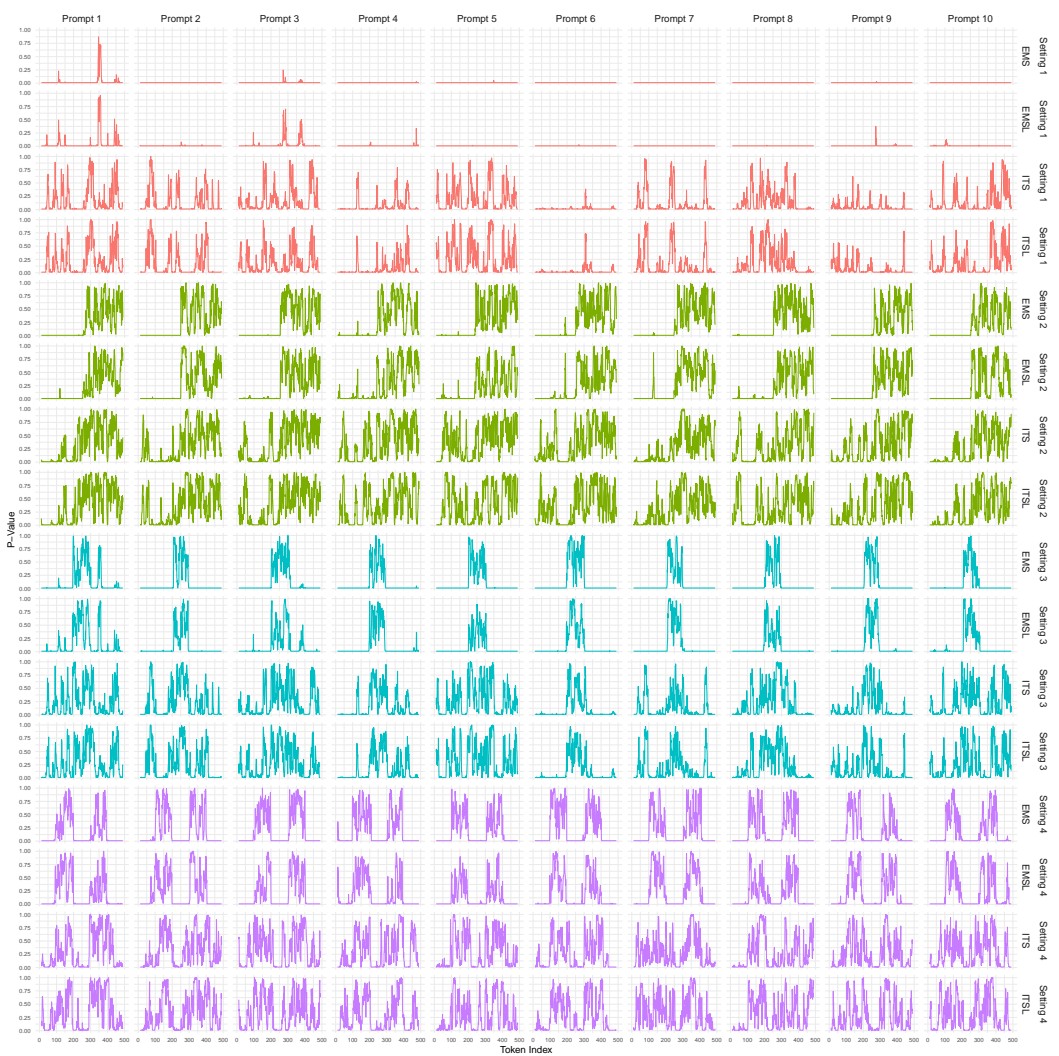

Figure B.3: Sequence of $p$-values for the first 10 prompts extracted from the Google C4 dataset for LLM `facebook/opt-1.3b`, organized into groups of four consecutive rows, each group corresponding to a distinct setting. Within each group, the rows represent $p$-values calculated using four different distance metrics: `EMS`, `EMSL`, `ITS`, and `ITSL`.

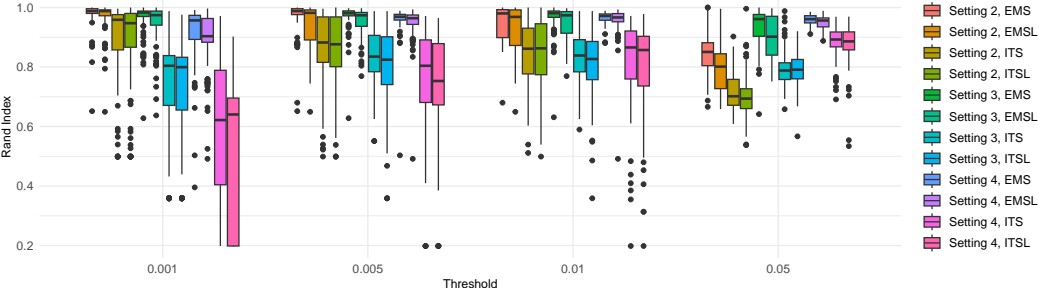

Figure B.4: The boxplots of the Rand index comparing the clusters identified through the detected change points for texts generated using `facebook/opt-1.3b` with the true clusters separated by the true change points with respect to different thresholds $\zeta$.

The sequence of $p$-values from the first 10 prompts extracted from the Google C4 dataset in all settings is presented in Figure B.3. The $p$-value sequences are organized into four groups, corresponding to the four settings in the numerical experiments section of the main text. Within each group, each row corresponds to one method.

By comparing with the $p$-value sequences obtained using `openai-community/gpt2`, we claim that the segmentation algorithm's performance depends on the quality of the $p$-values obtained for each sub-string and this relies on not only the watermark generation schemes but also the language models from which the texts are generated.

## B.4    Results for meta-llama/Meta-Llama-3-8B

Figure B.6 shows the results for false discoveries under Setting 1, where the text tokens are all watermarked and generated with `meta-llama/Meta-Llama-3-8B`. A lower threshold leads to fewer false discoveries.

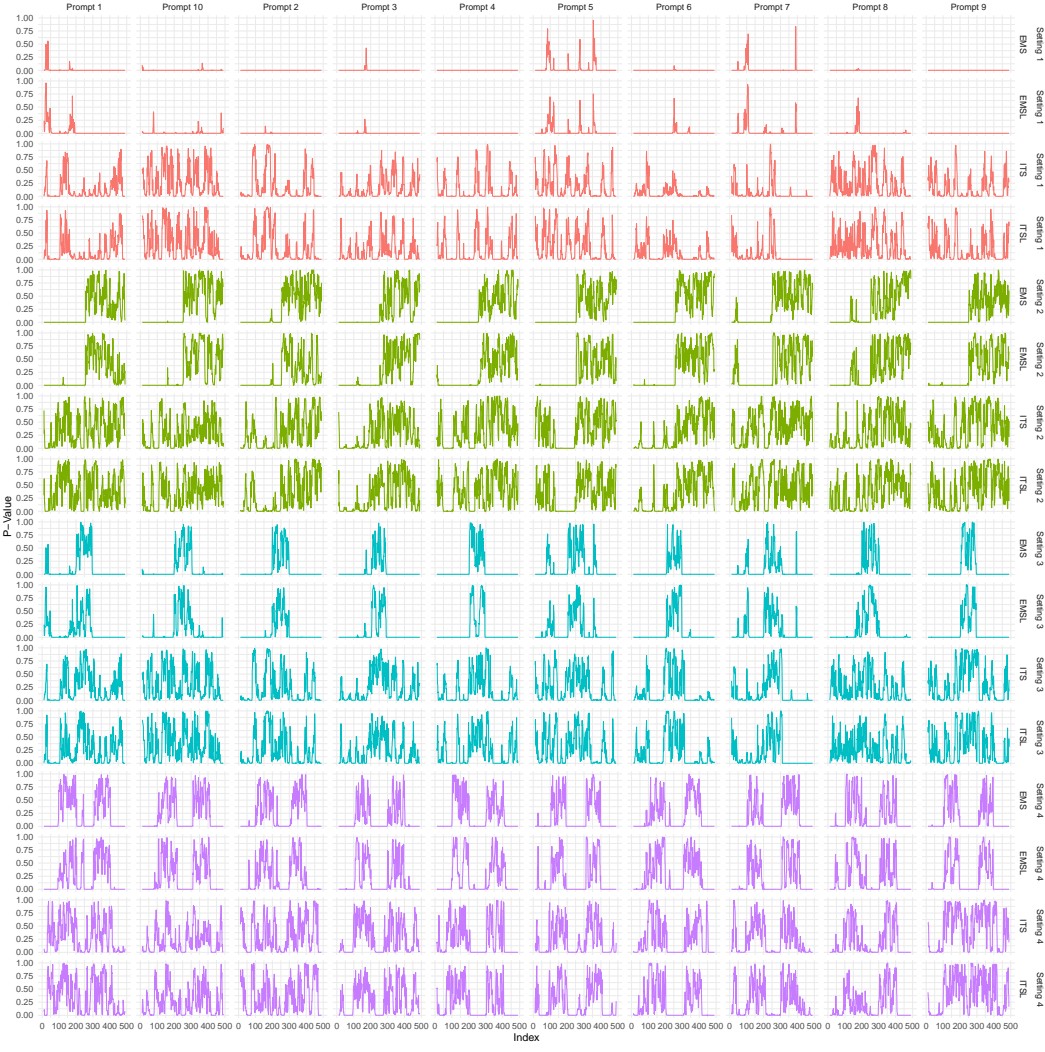

Figure B.5: Sequences of $p$-values for the first 10 prompts extracted from the Google C4 dataset for LLM `meta-llama/Meta-Llama-3-8B`, organized into groups of four consecutive rows, each group corresponding to a distinct setting. Within each group, the rows represent $p$-values calculated using four different distance metrics: EMS, EMSL, ITS, and ITSL.

Figure B.7 shows the boxplots of the Rand index for the four methods under different settings. EMS demonstrates the best performance across all settings, achieving its best performance at a threshold of $\zeta = 0.005$.

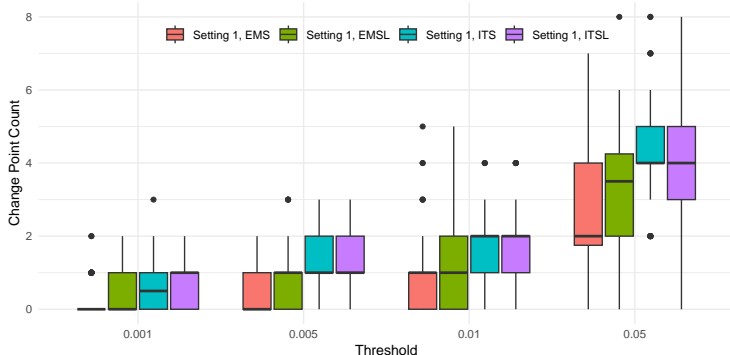

Figure B.6: The boxplots of the number of false positives with respect to different thresholds $\zeta$ under Setting 1. The texts are generated using `meta-llama/Meta-Llama-3-8B`, and the four distance metrics are EMS, EMSL, ITS, and ITSL.

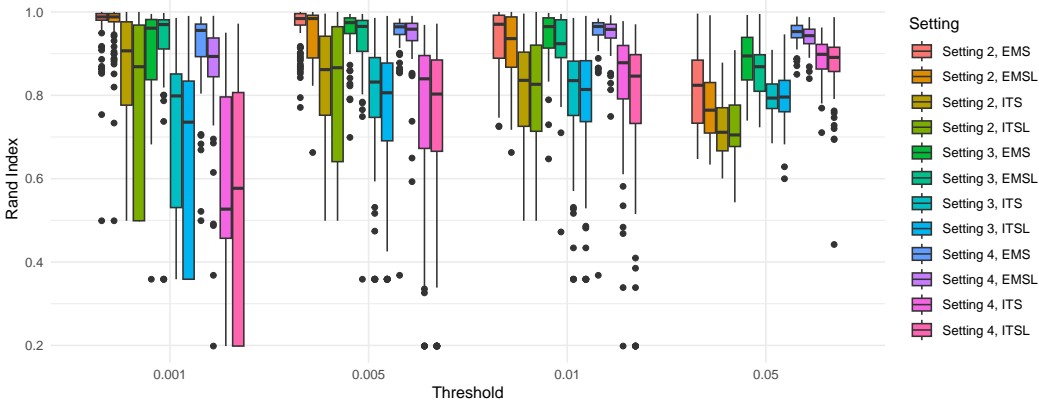

Figure B.7: The boxplots of the Rand index comparing the clusters identified through the detected change points for texts generated using `meta-llama/Meta-Llama-3-8B` with the true clusters separated by the true change points with respect to different thresholds $\zeta$.

## B.5    Other settings

We focus on the Meta-Llama-3-8B model to conduct simulation studies under some more difficult scenarios. Specifically, we increase the number of change points to 4, 8, and 12 and vary the segment lengths accordingly. The results are presented in Figure B.8. For scenarios with 4 and 8 change points, the proposed method successfully identifies all change points. As the number of change points increases, the change point detection problem becomes much more challenging. In a scenario with 12 change points, our method was able to identify 9 of them, showing its robust performance in handling more difficult situations. Generally, the difficulty of a change point detection problem depends on the distance between the change points and the magnitudes of the changes, as indicated by the theoretical results in Proposition 1.

## C    Choices of the tuning parameters

We investigate the impact of the window size $B$ and the number of permutations $T$ on the proposed method. In particular, we focus on the EMS method in simulation Setting 4 of the main text, using the Meta-Llama-3-8B model.

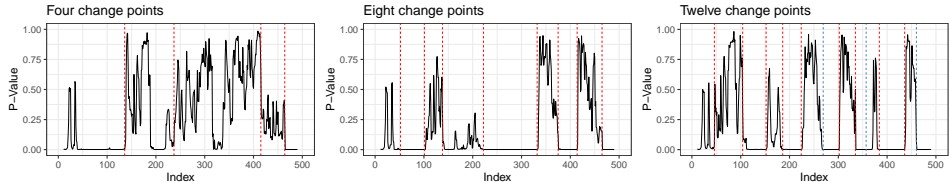

Figure B.8: Sequences of p-values obtained using the EMS method across various settings. Change points identified by the proposed method are marked with red dashed lines.

First, to study the impact of the window size $B$, we fix $T = 999$ and vary the value of $B$ in the set $\{10, 20, 30, 40, 50\}$. Table C.1 shows the rand index value for each setting, with a higher rand index indicating better performance.

Table C.1: Results for different choices of $B$ when $T = 999$.

| $B$ | 10 | 20 | 30 | 40 | 50 |
|---|---|---|---|---|---|
| Rand index | 0.8808 | 0.9429 | 0.9641 | 0.9570 | 0.9243 |

It is crucial to select an appropriate value for $B$. If $B$ is too small, the corresponding window may not contain enough data to reliably detect watermarks, as longer strings generally make the watermark more detectable. Conversely, if $B$ is excessively large, it might prematurely shift the detected change point locations, thus reducing the rand index. For instance, let us consider a scenario with 200 tokens where the first 100 tokens are non-watermarked, and the subsequent 100 are watermarked, with the true change point at index 101. Assuming our detection test is highly effective, then it will yield a p-value uniformly distributed over $[0, 1]$ over a non-watermarked window and a p-value around zero over a window containing watermarked tokens. When $B = 50$, the window beginning at the 76th token contains one watermarked token, which can lead to a small p-value and thus erroneously indicate a watermark from the 76th token onwards. In contrast, if $B = 20$, the window starting at the 91st token will contain the first watermarked token, leading to a smaller error in identifying the change point location. The above phenomenon is the so-called edge effect, which will diminish as $B$ gets smaller.

The trade-off in the choice of window size is recognized in the time series literature. For instance, a common recommendation for the window size in time series literature is to set $B = Cn^{1/3}$, where $n$ is the sample size, as seen in Corollary 1 of Lahiri [1999]. Based on our experience, setting $B = \lfloor 3n^{1/3} \rfloor$ (for example, when $n = 500$, $B = 23$) often results in good finite sample performance. A more thorough investigation of the choice of $B$ is deferred to future research.

We next examine how our method is affected by the choice of the number of permutations $T$. To this end, we set $B$ to 20 and consider $T \in \{99, 249, 499, 749, 999\}$. The results are summarized in Table C.2, including the rand index value and computation times for each setting. As expected, the computation time increases almost linearly with the number of permutations $T$. We also note that the rand index remains consistent across different values of $T$, indicating a level of stability in our method.

Table C.2: Results (computational time in hours and rand index values) for different choices of $T$ when $B = 20$.

| $T$ | 99 | 249 | 499 | 749 | 999 |
|---|---|---|---|---|---|
| Time in hours | 3.41 | 7.33 | 11.47 | 18.47 | 24.94 |
| Rand index | 0.937 | 0.9404 | 0.9326 | 0.9348 | 0.9354 |

