# OpenReview forum: "Segmenting Watermarked Texts From Language Models"
_NeurIPS.cc/2024/Conference — NeurIPS 2024 poster_

### Official Review · Reviewer_LBBH · 2024-07-08

**Soundness:** 3
**Presentation:** 3
**Contribution:** 3
**Rating:** 6
**Confidence:** 3

**Summary:**

The work makes two contributions: (1) A randomized-based test, including a theoretical analysis of a substring-based watermark detection for two popular distribution-preserving watermarks. The key idea is similar to the base detection methods in their respective base papers but shows some additional theoretical properties (e.g., convergence). (2) Based on the substring-based detection, an approach to segment the given text into both watermarked and non-watermarked sections by detecting points of change in the observed p-values on substrings. The proposed algorithm SeedBS-NOT is then evaluated in 4 settings and 2 models.

**Strengths:**

- The idea of differentiating between sections of text that are watermarked and the ones that are not is interesting and has many potential downstream applications (such as adaptive attacks).
- The theoretical analysis seems sound and does provide asymptotical guarantees that are new to the reviewer.
- In the evaluations, the change-points of watermarked and not-watermarked texts can be detected quite accurately.

**Weaknesses:**

- The method seems to rely heavily on the hyper-parameter B (substring block size) and would most likely profit from some ablation on the parameter in practical settings.
- The introduced scenarios for evaluation feel very clean, with large gaps between change points. However, I do not believe this accurately reflects real-world perturbations in the watermarked text.
- Similarly the evaluation is only one two quite oudated LLMs (gpt2 being 5 years old now) despite the paper claiming that the segementation algorithms performance "depends [on]  the language model with which the texts are generated". Given this I believe an extended evaluation on more realistic models (that people would actually watermark) is important to evaluate the empirical effectiveness of the approach (i.e., at least 7B -  the reviewer found in their own work that there can be significant changes between larger models when it comes to watermark detection).

## Minor

- The methods and analysis is restricted two specific types of watermarks, which are generally praised for their theoretical guarantees, but arguably not as popular as red-green type schemes for which such segmentations would also be interesting. Notably practical implementations of these schemes can, e.g., contain repeating key patterns whose effect would be interesting to investigate/mention in this context.
- Given this the results too me only feel partially encouraging in that we (despite theoretical asymptotic error bounds) do observe some false-positives in some of the settings (which qualitatively is confirmed by, e.g., Fig. B.1 where some of the results tend to be quite noisy especially for ITS(L)).

## Typos/Nits

- L.27 Large Language Model
- L.291 400 -> 500. According to the plots and the fact that there are 4 change-points it should be 500 tokens (same in L. 579)

**Questions:**

- How does the proposed method fare on more realistic text scenarios with shorter sections of watermarked / non-watermarked texts?
- Can you provide a practical ablation over B (both w.r.t. runtime and effectiveness of the detection)?
- Can you provide some experiments on larger and more realistic models (As well as the costs associated with running tests on such models).
- Do you have any ideas about the effect of repeating key-patterns (which happens in practice) on your analysis?

**Limitations:**

The paper addresses several limitations; however, as pointed out above, I think some more limitations of the current setting could be discussed.

---

> ### Author Rebuttal · Authors · 2024-08-07
>
> Thank you for taking the time to review our work. We will provide a detailed response to each of your comments below.
>
> ### **Weaknesses and Questions:**
>
> *1. The method seems to rely heavily on the hyper-parameter $B$ and would most likely profit from some ablation on the parameter in practical settings. Can you provide a practical ablation over $B$?*
>
> **Ans:** Thanks for the comment. We investigate the impact of the window size $B$ on the proposed method. In particular, we focus on the EMS method in simulation Setting 4 of the main text, using the Meta-Llama-3-8B model. To study the impact of the window size $B$, we fix $T=999$ and vary the value of $B$ in the set $\{10, 20, 30, 40, 50\}$. Below are the computation times and the rand index value for each setting, with a higher rand index indicating better performance.
>
>    | $B$ | 10 | 20 | 30 | 40 | 50 |
>    |-------|----|----|----|----|----|
>    | rand index | 0.8808 | 0.9429 | 0.9641 | 0.9570 | 0.9243 |
>    | time in hours | 28.77 | 24.94 | 24.37 | 26.24 | 25.97 |
>
> It is crucial to select an appropriate value for $B$. If $B$ is too small, the corresponding window may not contain enough data to reliably detect watermarks, as longer strings generally make the watermark more detectable. Conversely, if $B$ is excessively large, it might prematurely shift the detected change point locations, thus reducing the rand index. Please refer to Author Rebuttal point 2 for one example.
>
> The trade-off in the choice of window size is recognized in the time series literature. For instance, a common recommendation for the window size in time series literature is to set $B=Cn^{1/3}$, where $n$ is the sample size, as seen in Corollary 1 of Lahiri (1999). Based on our experience, setting $B = \lfloor 3n^{1/3} \rfloor$ often results in good finite sample performance. A more thorough investigation of the choice of $B$ is deferred to future research. Additionally, The selection of $B$ does not significantly affect the time.
>
> *2. The introduced scenarios for evaluation feel very clean, with large gaps between change points. However, I do not believe this accurately reflects real-world perturbations in the watermarked text. How does the proposed method fare on more realistic text scenarios with shorter sections of watermarked / non-watermarked texts?*
>
> **Ans:** Thank you for your comment. We focus on the Meta-Llama-3-8B model to conduct more realistic simulation studies. Specifically, we increase the number of change points to 4, 8, and 12 and vary the segment lengths accordingly. The results are presented in the attached pdf file Figure 2. For scenarios with 4 and 8 change points, the proposed method successfully identifies all change points. As the number of change points increases, the change point detection problem becomes much more challenging. In the scenario with 12 change points, our method was able to identify 9 of them, showing its robust performance in handling more difficult situations. Generally, the difficulty of a change point detection problem depends on the distance between the change points and the magnitudes of the changes, as indicated by the theoretical results in Proposition 1.
>
> *3. I believe an extended evaluation on more realistic models is important to evaluate the empirical effectiveness of the approach. Can you provide some experiments on larger and more realistic models?*
>
> **Ans:** Thanks for the comment. We conducted simulation tests using the Llama model, specifically the Meta-Llama-3-8B, which was released on April 18, 2024, and has 8 billion parameters. Our experiments were carried out under three different settings outlined in the main text: no change points, one change point, and four change points. The results, displayed in the attached pdf file Figure 1, demonstrate the robust performance of the proposed method under the Meta-Llama-3-8B model. We found that the p-values are accurate (e.g., small for watermarked segments), and this precision in the p-value calculation leads to accurate change point detection, confirming the effectiveness of the proposed method in handling complex LLM.
>
> ### **Minor**
>
> *1. The methods [...] on your analysis?*
>
> **Ans:** Thank you for the excellent suggestion! We can indeed combine the red-green type scheme with the proposed change point detection algorithm, as outlined below. The proposed method operates in two stages. First, a watermark detection method is used to produce a sequence of p-values. These p-values are then utilized for change point detection. Therefore, any watermark detection method that produces a sequence of p-values can be integrated into our framework to differentiate between watermarked and non-watermarked text. As red-green type schemes can generate p-values, they can be included in the proposed framework to segment watermarked texts. We are currently exploring this extension and will report the numerical results in the revision.
>
> *2. Given this [...] the settings.*
>
> **Ans:** The false discoveries are partially due to the low detection power (i.e., large p-values for watermarked segments). In reality, the detection power depends crucially on the text itself (such as its length and entropy). The text's characteristics can determine how challenging it is to detect the watermarks. Our theory suggests that the detection power will only approach one when the quantity defined in Equation (4) (related to the text's entropy) approaches infinity. For a given text, if this quantity is small, it would become difficult to find the watermark.
>
> ### **Typos/Nits**
>
> *1. L.27 Large Language Model*
>
> **Ans:** Thanks for catching this. It will be fixed.
>
> *2. L.291 400 -> 500. [...] (same in L. 579)*
>
> **Ans:** This is not a typo. The total number of tokens is indeed 500, where we first generate 400 tokens with watermark, and then insert an additional 100 tokens.
>
> ###  References
> [1] Soumendra N Lahiri. Theoretical comparisons of block bootstrap methods. Annals of Statistics, pages 386–404, 1999

---

> > ### Comment · Reviewer_LBBH · 2024-08-08
> >
> > I thank the author for their extensive rebuttal and appreciate the time and effort spent on new experiments. The presented results seem convincing and address most of my main concerns (I remain very interested in adaptations to other watermarking schemes). In light of the new results, I will raise my score.
> >
> > P.S.: I must have glossed over the inserted in L.293 multiple times; this is clearly on me.

---

> ### Author Response · Authors · 2024-08-09
>
> Thank you for your thoughtful and constructive feedback. We are grateful for your decision to raise your score and your efforts in enhancing our manuscript. We are thrilled to hear that the additional experiments and results have addressed most of your concerns. We will definitely consider additional watermark schemes (e.g., the red-green scheme with possibilities of key-repeating patterns) in our revision if time permits and in our future work.

---

### Official Review · Reviewer_XFsq · 2024-07-08

**Soundness:** 3
**Presentation:** 2
**Contribution:** 3
**Rating:** 5
**Confidence:** 2

**Summary:**

This paper presents a statistical method for detecting and segmenting watermarked text generated by large language models (LLMs). The key contributions are:

A rigorous analysis of Type I and Type II errors for a randomization test to detect the presence of watermarks in generated text. The authors apply their findings to two specific watermarking schemes: inverse transform sampling and exponential minimum sampling (Gumbel watermark).
A novel statistical approach to segment text into watermarked and non-watermarked substrings. This method is based on change point detection techniques and is designed to handle scenarios where users may modify LLM-generated text through insertions, deletions, or substitutions.
Theoretical analysis of the proposed segmentation method, including convergence rates for estimated change point locations.
Empirical evaluation of the proposed methods using texts generated by GPT-2 and OPT-1.3b models, with prompts from Google's C4 dataset. The experiments demonstrate the effectiveness of the approach under various modification scenarios.

The paper provides a theoretical foundation for watermark detection and segmentation in LLM-generated text, addressing an important problem in the context of distinguishing between human-written and machine-generated content.

**Strengths:**

Originality:

A novel statistical approach for watermarked text segmentation, is very interesting and useful.

Creative application of change point detection to watermarking.

Quality:

Rigorous theoretical analysis with proofs.

Thorough experimental design across multiple scenarios.

Clarity:

Well-structured paper with clear progression.

Readable mathematical concepts with intuitive explanations.

Significance:

Addresses key problem of distinguishing AI-generated text in "segmented text" scenario.

**Weaknesses:**

Weaknesses:

Limited model diversity: Experiments focus on GPT-2 and OPT-1.3b. Testing on larger language models would strengthen findings.

While I appreciate the intent of this paper, the attack scenario set up in this article is still rather simple. In a real-world application,
1) Users may make changes to the text by means of cross-lingual attack and GPT rewrite, etc., the authors did not analyze these scenarios.
2) The precautions users may take to prevent possible watermarking are more complex. I would like to see more change points and performance ratings of methods for scenarios with different segment lengths.

In addition, better visualization for selected scenarios could better reflect the value of the accuracy of the proposed methodology.

It should be noted that I will raise my score when I see appropriate rebuttals.

**Questions:**

1. I am interested in the number of $T$ selected for watermarked text detection. Is there a solution for balancing performance and speed?

**Limitations:**

The authors have adequately addressed the limitations in the checklist.

---

> ### Author Rebuttal · Authors · 2024-08-07
>
> We appreciate your comments and we provide a point-by-point response to each of your comments below.
>
> *1. Limited model diversity [...] larger language models would strengthen findings.*
>
> **Ans:** Thanks for the comment. We conducted simulation tests using the Llama model, specifically the Meta-Llama-3-8B, which was released on April 18, 2024, and has 8 billion parameters. Our experiments were carried out under three different settings outlined in the main text: no change points, one change point, and four change points. The results, displayed in the attached PDF file Figure 1, demonstrate the robust performance of the proposed method under the Meta-Llama-3-8B model. We found that the p-values are accurate (e.g., small for watermarked segments), and this precision in the p-value calculation leads to accurate change point detection, confirming the effectiveness of the proposed method in handling complex LLM.
>
> *2. Feedback on Scenario Simplicity: While I appreciate the intent of this paper, the attack scenario set up in this article is still rather simple. In a real-world application, users may make changes to the text by means of cross-lingual attack and GPT rewrite, etc.; the authors did not analyze these scenarios.*
>
> **Ans:** Thank you for the feedback. During the response period, we tested various realistic scenarios, including cross-lingual attacks and rewriting using different Language Models (LLMs).
>
> Regarding the use of different LLMs for rewriting, we would like to note the following: If the entire text is rewritten using another LLM and the associated key is provided, we can use the method outlined in the paper to detect watermarks. However, if the key from the alternate LLM used for rewriting is not provided, it becomes difficult to detect such an attack. In a situation where a portion of the text is rewritten using a different LLM (for example, LLM-B compared to the original LLM, denoted as LLM-A used for generating the watermarked text) and the corresponding watermark key is provided, we can address the issue by applying the change-point detection algorithm separately to each of the LLMs and then consolidating the identified change points.
>
> To test the rewrite attack, we first generated 300 tokens using Meta-Llama-3-8B (LLM-A), with the initial 100 tokens unwatermarked and the subsequent 200 tokens watermarked. Subsequently, the openai-community/gpt2 (LLM-B) revised the text corresponding to the final 100 tokens generated by LLM-A. We then employed the proposed change point detection method using LLM-A and LLM-B, separately. The results, illustrated in Figure 3 of the attached PDF, indicate that LLM-A exhibits a p-value near zero between tokens $100$ and $200$, while LLM-B shows a near-zero p-value between tokens $200$ and $300$. This evidence supports the efficacy of the proposed method in distinguishing text generated by two different LLMs.
>
> We also conducted tests for cross-lingual attacks. Under Setting 4 detailed in the main text, we translated the paragraph into French and then back into English. The results are shown in Figure 4 of the attached PDF. The change points are easily detected before translation. However, after translation, the sequence of $p$-values becomes noisy, which significantly degrades the performance of change point detection. The noisiness of the $p$-values is due to potential alterations in word order or logic during the translation process.
>
> *3. Precautions Against Watermarking: [...] more change points and performance ratings of methods for scenarios with different segment lengths.*
>
> **Ans:** Thanks for the comments. We focus on the Meta-Llama-3-8B model to conduct more realistic simulation studies. Specifically, we increase the number of change points to $4$, $8$, and $12$ and vary the segment lengths accordingly. The results are presented in the attached PDF file Figure 2. For the scenarios with $4$ and $8$ change points, the proposed method successfully identifies all change points. As the number of change points increases, the change point detection problem becomes much more challenging. In the scenario with 12 change points, our method was able to identify 9 of them, showing its robust performance in handling more difficult situations.
>
> *4. Visualization Enhancement: In addition, better visualization for selected scenarios [...].*
>
> **Ans:** Thank you for your comment. To demonstrate our approach more concretely, we will provide an example of segmenting a specific string. Due to the space constraint, we have relocated the detailed text to Figure 5 in the attached PDF. In this figure, true watermarked text highlighted in green and detected watermarked text highlighted in blue.
>
> *5. Watermarked Text Detection: I am interested in the number of $T$ selected for watermarked text detection. Is there a solution for balancing performance and speed?*
>
> **Ans:** In theory, a larger $T$ is preferable as it leads to a more accurate calculation of the p-values. Following the literature on permutation-based tests (Marozzi, 2004), we suggest using 500-1000 permutations. We investigate the impact of the number of permutations $T$ on the proposed method, where we focus on the EMS method in simulation Setting 4, using the Meta-Llama-3-8B model. We set $B$ to 20 and consider $T\in \{99, 249, 499, 749, 999\}$. The results are summarized below. As expected, the computation time increases almost linearly with the number of permutations $T$. We also note that the rand index remains consistent across different values of $T$, indicating a level of stability in our method.
>
> | $T$  | 99   | 249   | 499   | 749   | 999   |
> |------|------|-------|-------|-------|-------|
> | time in hours | 3.41 | 7.33  | 11.47 | 18.47 | 24.94 |
> | rand index    | 0.937 | 0.9404 | 0.9326 | 0.9348 | 0.9354 |
>
> ###  References
> [1] Marco Marozzi. Some remarks about the number of permutations one should consider to perform a permutation test. Statistica, 64(1):193–201, 2004.

---

> > ### Comment · Reviewer_XFsq · 2024-08-08
> >
> > Thank you for addressing my concerns and providing detailed experimental results. I appreciate the additional experiments conducted with the *Meta-Llama-3-8B* model and the analyses of cross-lingual attacks and LLM rewrites. The improved visualizations and theoretical insights significantly strengthen your paper.
> >
> > However, as mentioned in my initial review, I still did not see the results incorporating multiple change points with varying segment lengths. For instance, an experiment with the following token structure would be highly insightful:
> >
> > - Tokens 1-50: Unwatermarked
> > - Tokens 50-150: Watermarked
> > - Tokens 150-400: Unwatermarked
> > - Tokens 400-650: Watermarked
> > - Tokens 650-950: Unwatermarked
> > - Tokens 950-1300: Watermarked
> >
> > Such scenarios would better reflect real-world complexities and enhance the robustness of your methodology.
> >
> > Thank you again for your thorough responses and valuable additions to the paper.

---

> ### Author Response · Authors · 2024-08-09
>
> We appreciate your comments, which have significantly helped us improve the manuscript. Also, thank you for taking the time to review our rebuttal once more. In response to your comment, "The precautions users [...] different segment lengths," we have indeed conducted numerical experiments with varying segment lengths and have presented the results in Figure 2 of the attached PDF file. We apologize for not making the setups clear in our rebuttal. Below, we clarify the simulation settings considered in Figure 2.
>
> Specifically, we generated change point locations randomly. For instance, in the most challenging scenario with 12 change points, the true change points are located at {46, 113, 151, 172, 222, 269, 297, 336, 357, 382, 425, 460}. More precisely, we generate the texts such that
> - Tokens 1-45: Watermarked
> - Tokens 46-112: Unwatermarked
> - Tokens 113-150: Watermarked
> - Tokens 151-171: Unwatermarked
> - Tokens 172-221: Watermarked
> - Tokens 222-268: Unwatermarked
> - Tokens 269-296: Watermarked
> - Tokens 297-335: Unwatermarked
> - Tokens 336-356: Watermarked
> - Tokens 357-381: Unwatermarked
> - Tokens 382-424: Watermarked
> - Tokens 425-459: Unwatermarked
> - Tokens 460-500: Watermarked
>
> This setup is challenging due to the varying segment lengths and the small gaps between the change point locations. In this case, the set of change points detected by our algorithm is given by {46, 104, 152, 186, 224, 302, 335, 384, 436}, which successfully identifies 9 of the 12 change points. In the scenario with 8 change points with random locations, our method was able to detect all 8 change point locations successfully. These results demonstrate the robustness/effectiveness of our proposed method in challenging situations.
>
> Based on the setting you suggested, we have conducted an additional experiment. Specifically, we generate texts with 1300 tokens and the true change points located at {51, 151, 401, 651, 951}, based on the substitution attacks. The other configurations are the same as in the rebuttal. The detected change points are {57, 154, 409, 667, 955}. We consider this result promising, and if time permits, we shall obtain more experimental results in the next few days.

---

> > ### Comment · Reviewer_XFsq · 2024-08-10
> >
> > Thank you for your detailed response and the additional experiments. I appreciate the efforts you've put in during the rebuttal period.
> >
> > I apologize for overlooking the details in Figure 2 earlier. I now recognize that it does address the scenario of multiple change points, though the segment lengths were not explicitly pointed out.
> >
> > Your thorough rebuttal has addressed most of my concerns. I am particularly impressed by the additional experiments, especially the one following the token structure I suggested. These results demonstrate the robustness of your methodology.
> >
> > Given your responses and the considerable effort in improving the paper, I am inclined to raise my evaluation positively. Thank you again for your hard work.

---

> > > ### Author Response · Authors · 2024-08-11
> > >
> > > Thank you for your encouraging feedback and for acknowledging the additional experiments. We are pleased that our efforts have effectively addressed your concerns, and we are glad to share the updated Rand Index results obtained from 10 texts with 1300 tokens:
> > >
> > > | Threshold |  0.05  |  0.01  |  0.005 |  0.001 |
> > > |-----------|--------|--------|--------|--------|
> > > | Rand Index| 0.9489 | 0.9523 | 0.9553 | 0.9767 |
> > >
> > > The threshold refers to the $p$-value threshold in NOT used to claim the significance of a change point. A higher Rand Index indicates better performance. These results are promising and demonstrate the robustness of our proposed method.

---

### Official Review · Reviewer_SyZV · 2024-07-10

**Soundness:** 4
**Presentation:** 4
**Contribution:** 3
**Rating:** 7
**Confidence:** 4

**Summary:**

This paper considers the problem of segmenting a watermarked text into watermarked and unwatermarked subsequences. This is achieved using change point detection methods. There is a nice statistical analysis of the single change point detection problem, and a heuristic algorithm (from Kovács et al., 2022) with experimental results for multiple change point detection.

**Strengths:**

The paper is easy to read.

Section 2 (Problem setup) presents a result (Theorem 1) on the power of the watermark test that is more rigorous than previous presentations (to my knowledge).

The connection to change-point detection literature with analysis using the block-bootstrap is interesting (Section 3). While I am not deeply familiar with the change-point literature, the connections and application of these methods appears correct and well-cited.

The experiments appear well-executed.

**Weaknesses:**

This paper could be criticized for novelty, in the sense that it mostly applies standard tools from the change point detection literature to watermarking. I find Remark 2 somewhat cryptic; perhaps it speaks to some novelty of the analysis in Section 3, in which case this should be clarified.

Unbiased watermarks work best for high entropy text; small models like gpt2/opt-1.3b tend to produce high-entropy text. It would be informative to see whether change-point detection remains practical for the text generated by larger models (e.g., Mistral, LLaMA).

**Questions:**

Why was block size B=20 chosen? Is it important that B = B'? How do these and other parameters affect the performance of SeedBS in this setting?

**Limitations:**

Limitations are adequately addressed.

---

> ### Author Rebuttal · Authors · 2024-08-06
>
> Thank you for taking the time to review our work. We greatly appreciate your positive feedback. We provide a detailed response to each of your comments below.
>
> *1. This paper could be criticized for novelty, in the sense that it mostly applies standard tools from the change point detection literature to watermarking. I find Remark 2 somewhat cryptic; perhaps it speaks to some novelty of the analysis in Section 3, in which case this should be clarified.*
>
> **Ans:** Thanks for the comment. We will update Remark 2 to highlight the novelty of using the change point detection technique in the current context. Specifically, we shall emphasize the following points:
> - Rather than testing the homogeneity of the original data sequence (which is the setup typically considered in the change point literature), we convert the string into a sequence of p-values, based on which we conduct the change-point analysis;
> - In the classical change point literature, the observations (in our case, the p-values) within the same segment are assumed to follow the same distribution. In contrast, for the watermark detection problem, the p-values from the watermarked segment could follow different distributions, adding a layer of difficulty to the analysis;
> - The p-value sequence is dependent (where the strength of dependence is controlled by $B$), making our setup very different from the one in Carlstein (1988), which assumed the underlying data sequence to be independent;
> - The technical tool used in our analysis must account for the particular dependence structure within the p-value sequence, which is also different from existing works in the literature.
>
> *2. Unbiased watermarks work best for high entropy text; small models like gpt2/opt-1.3b tend to produce high-entropy text. It would be informative to see whether change-point detection remains practical for the text generated by larger models (e.g., Mistral, LLaMA).*
>
> **Ans:** Following your suggestion, we have conducted simulation tests using the Llama model, specifically the Meta-Llama-3-8B, which was released on April 18, 2024, and has 8 billion parameters. Our experiments were carried out under three different settings outlined in the main text: no change points, one change point, and four change points. The results, displayed in the attached PDF file Figure 1, demonstrate the robust performance of the proposed method under the Meta-Llama-3-8B model. We found that the p-values are accurate (e.g., small for watermarked segments), and this precision in the p-value calculation leads to accurate change point detection, confirming the effectiveness of the proposed method in handling complex LLM.
>
> *3. Why was block size $B=20$ chosen? Is it important that $B = B'$? How do these and other parameters affect the performance of SeedBS in this setting?*
>
> **Ans:** Thank you for your comment. Following your suggestion, we investigate the impact of the window size $B$ on the proposed method. In particular, we focus on the EMS method in simulation Setting 4 of the main text, using the Meta-Llama-3-8B model.
>
> To study the impact of the window size $B$, we fix $T=999$ and $B' = 20$, and vary the value of $B$ in the set \{10, 20, 30, 40, 50\}. The results for different choices of $B$ when $T=999$ are shown below:
>
> | $B$ | 10 | 20 | 30 | 40 | 50 |
> |-------|----|----|----|----|----|
> | rand index | 0.8808 | 0.9429 |  0.9641 | 0.9570 |0.9243 |
>
> It is crucial to select an appropriate value for $B$. If $B$ is too small, the corresponding window may not contain enough data to reliably detect watermarks, as longer strings generally make the watermark more detectable. Conversely, if $B$ is excessively large, it might prematurely shift the detected change point locations, thus reducing the rand index.
>
> The trade-off in the choice of window size is recognized in the time series literature. For instance, a common recommendation for the window size in time series literature is to set $B=Cn^{1/3}$, where $n$ is the sample size, as seen in Corollary 1 of  Lahiri (1999). Based on our experience, setting $B = \lfloor 3n^{1/3} \rfloor$ (for example, when $n=500$, $B=23$) often results in good finite sample performance. A more thorough investigation of the choice of $B$ is deferred to future research.
>
> Our experimental results indicate that setting $B = B'$ is unnecessary. In practice, the sequence of $p$-values is $B$-dependent, with $p_i$ and $p_j$ being independent only if $|i - j| > B$. Consequently, we recommend using $B' = B$ to ensure that the block bootstrap adequately captures this dependence.
>
> ###  References
> [1] Edward Carlstein. Nonparametric change-point estimation. The Annals of Statistics, 16(1):188–197, 1988\
> [2] Soumendra N Lahiri. Theoretical comparisons of block bootstrap methods. Annals of Statistics, pages 386–404, 1999

---

> > ### Comment · Reviewer_SyZV · 2024-08-10
> >
> > Thank you for your thorough response.
> >
> > All of my concerns have been addressed: I have raised my score to reflect this. This is a good paper and I hope it is accepted.

---

> > > ### Author Response · Authors · 2024-08-11
> > >
> > > Thank you very much for your positive feedback and for raising your score. We are grateful for your thorough review and are delighted to hear that our response has addressed all your concerns.

---

### Author Rebuttal · Authors · 2024-08-06

First and foremost, we express our sincere gratitude to the three reviewers for their invaluable feedback, which has helped us improve the quality of our manuscript. We will revise our manuscript and the supplement, taking all the reviewers' comments into consideration. Before providing point-by-point responses, we would like to summarize the major changes we have made in order to address the reviewers' main concerns.

### 1. **Larger and more realistic/updated LLMs**
We conducted simulation tests using the Llama model, specifically the Meta-Llama-3-8B, which was released on April 18, 2024, and has 8 billion parameters. Our experiments were carried out under three different settings outlined in the main text: no change points, one change point, and four change points. The results, displayed in the attached pdf file Figure 1, demonstrate the robust performance of the proposed method under the Meta-Llama-3-8B model. We found that the p-values are accurate (e.g., small for watermarked segments), and this precision in the p-value calculation leads to accurate change point detection, confirming the effectiveness of the proposed method in handling complex LLM.

### 2. **More realistic evaluation scenarios**
We focus on the Meta-Llama-3-8B model to conduct more realistic simulation studies. Specifically, we increase the number of change points to 4, 8, and 12 and vary the segment lengths accordingly. The results are presented in the attached pdf file Figure 2. For scenarios with 4 and 8 change points, the proposed method successfully identifies all change points. As the number of change points increases, the change point detection problem becomes much more challenging. In the scenario with 12 change points, our method was able to identify 9 of them, showing its robust performance in handling more difficult situations. Generally, the difficulty of a change point detection problem depends on the distance between the change points and the magnitudes of the changes, as indicated by the theoretical results in Proposition 1.

### 3. **Choices of the tuning parameters**
We investigate the impact of the window size $B$ and the number of permutations $T$ on the proposed method. In particular, we focus on the EMS method in simulation Setting 4 of the main text, using the Meta-Llama-3-8B model.

First, to study the impact of the window size $B$, we fix $T=999$ and vary the value of $B$ in the set \{10, 20, 30, 40, 50\}. The rand index values for each setting are outlined below, with a higher rand index indicating better performance:

| $B$  | 10     | 20     | 30     | 40     | 50     |
|------|--------|--------|--------|--------|--------|
| rand index | 0.8808 | 0.9429 | 0.9641 | 0.9570 | 0.9243 |

In practice, it is crucial to select an appropriate value for $B$. If $B$ is too small, the corresponding window may not contain enough data to reliably detect watermarks, as longer strings generally make the watermark more detectable. Conversely, if $B$ is excessively large, it might shift the detected change point locations, thus reducing the rand index. For instance, let us consider a scenario with 200 tokens where the first 100 tokens are non-watermarked, and the subsequent 100 are watermarked, with the true change point at index 101. Assuming our detection test is valid in size and highly effective in detection power, then it will yield a p-value uniformly distributed over $[0,1]$ over a non-watermarked window and a p-value around zero over a window containing watermarked tokens. When $B = 50$, the window beginning at the 76th token contains one watermarked token, which can lead to a small p-value and thus erroneously indicate a watermark from the 76th token onwards. In contrast, if $B = 20$, the window starting at the 91st token will contain the first watermarked token, leading to a more minor error in identifying the change point location. The above phenomenon is the so-called edge effect, which will diminish as $B$ gets smaller.

The trade-off in the choice of window size is well recognized in the time series literature. For instance, a common recommendation for the window size in time series literature is to set $B=Cn^{1/3}$, where $n$ is the sample size, as seen in Corollary 1 of  Lahiri (1999). Based on our experience, setting $B = \lfloor 3n^{1/3} \rfloor$ (for example, when $n=500$, $B=23$) often results in good finite sample performance. A more thorough investigation of the choice of $B$ is deferred to future research.

We next examine how our method is affected by the choice of the number of permutations $T$. The results are summarized below, including the rand index value and computation times for each setting. As expected, the computation time increases almost linearly with the number of permutations $T$. We also note that the rand index remains consistent across different values of $T$, indicating a level of stability in our method.

| $T$  | 99   | 249   | 499   | 749   | 999   |
|------|------|-------|-------|-------|-------|
| time in hours | 3.41 | 7.33  | 11.47 | 18.47 | 24.94 |
| rand index    | 0.937 | 0.9404 | 0.9326 | 0.9348 | 0.9354 |

###  References
[1] Soumendra N Lahiri. Theoretical comparisons of block bootstrap methods. Annals of Statistics, pages 386–404, 1999

---

### Decision · Program_Chairs · 2024-09-25

**Decision:**

Accept (poster)

**Comment:**

This paper presents a statistical method for detecting and segmenting watermarked and unwatermarked text generated by large language models (LLMs).

Contributions and Strengths:
1. The paper presents a theoretical analysis of Type I and Type II errors for a randomization test to detect the presence of watermarks in generated text. They also examined two watermarking methods.
2. The paper proposes a statistical detection method to segment text into watermarked and non-watermarked text.
3. The paper provides a theoretical analysis of the proposed segmentation method.

Weakness:
1. The experiments are a bit weak. They just studied GPT-2 and OPT1.3B models, which are quite old and small.
2. The method heavily relies on the hyperparameter B.

Overall, the paper makes an important study on LLM watermarking and would be beneficial to the community.